# Magnesium galvanic cells produce hydrogen and modulate the tumor microenvironment to inhibit cancer growth

Nailin Yang[1], Fei Gong[1], Bo Liu[1], Yu Hao[1], Yu Chao[1], Huali Lei[1], Xiaoyuan Yang[1], Yuehan Gong[1], Xianwen Wang [1], Zhuang Liu [1✉] & Liang Cheng [1✉]

Hydrogen can be used as an anti-cancer treatment. However, the continuous generation of $H_2$ molecules within the tumor is challenging. Magnesium (Mg) and its alloys have been extensively used in the clinic as implantable metals. Here we develop, by decorating platinum on the surface of Mg rods, a Mg-based galvanic cell (MgG), which allows the continuous generation of $H_2$ in an aqueous environment due to galvanic-cell-accelerated water etching of Mg. By implanting MgG rods into a tumor, $H_2$ molecules can be generated within the tumor, which induces mitochondrial dysfunction and intracellular redox homeostasis destruction. Meanwhile, the $Mg(OH)_2$ residue can neutralize the acidic tumor microenvironment (TME). Such MgG rods with the micro-galvanic cell structure enable hydrogen therapy to inhibit the growth of tumors, including murine tumor models, patient-derived xenografts (PDX), as well as $VX_2$ tumors in rabbits. Our research suggests that the galvanic cells for hydrogen therapy based on implantable metals may be a safe and effective cancer treatment.

[1] Institute of Functional Nano & Soft Materials (FUNSOM), Jiangsu Key Laboratory for Carbon-Based Functional Materials and Devices, Soochow University, 215123 Suzhou, China. ✉email: zliu@suda.edu.cn; lcheng2@suda.edu.cn

As an emerging therapeutic strategy, gas therapy has attracted wide attention for cancer treatment[1–4]. Among the numerous therapeutic gases (e.g., CO, NO, $H_2S$, etc.), $H_2$ gas has a significant safety benefit[5,6]. $H_2$ gas is currently being used to treat a variety of ailments, including cancer, type 2 diabetes, Parkinson's disease, Alzheimer's disease, stroke, and arthritis[7,8]. As a reductive gas, the anti-inflammatory mechanism of $H_2$ has been widely established, while the specific mechanism of the $H_2$-induced antitumor action remains to be further understood[9–11]. It has been found that a low concentration of $H_2$ could regulate the inflammatory, while $H_2$ at a high concentration would inhibit cell mitochondrial respiration and break redox homeostasis, thus causing cancer cell apoptosis and damages[2,5,12,13]. The first example of $H_2$ therapy was using hyperbaric $H_2$ to treat cutaneous squamous cell carcinoma, presenting a great tumor-suppressive effect[14]. In recent years, $H_2$ therapy was mainly focused on nanomedicine strategies to deliver $H_2$ into the tumor, or generate $H_2$ within the tumor in response to endogenous or external stimulations[15–18]. For instance, palladium hydride ($PdH_{0.2}$) nanocrystals, $SnS_{1.68}$-$WO_{2.41}$ nanocatalysts, and Au-$TiO_2$ heterojunction nanoplatforms (Au-$TiO_2$@ZnS) have been used to realize in situ $H_2$ release for tumor gas therapy. However, achieving significant and long-term $H_2$ release in the tumor to produce the best anticancer therapeutic impact was still problematic. Therefore, for future clinical translation of $H_2$ therapy, alternative relevant ways to achieve continuous and efficient $H_2$ generation within the tumor in a safe manner are needed.

The reaction between reductive metals and $H_2O$ could generate high-purity $H_2$ gas[19,20]. Such a strategy has rarely been applied in biomedicine except in a recent paper reporting the use of Fe needle electrodes to react with $H_2O$ and generate $H_2$ for cancer therapy under an external electric power supply[12]. However, the requirement of an external power supply might make it less practical for clinical application in the future. In contrast, magnesium (Mg) and its alloys have been widely used as implant metals in clinic[21–24]. Recently, implantable materials and devices (e.g., radioactive seeds) have been widely used in cancer treatments (e.g., liver cancer, and prostate cancer) and have achieved great therapeutic effects. Moreover, the local implantation could avoid the systemic toxicity of therapeutic agents[25,26]. Local administration of the implantable $H_2$ therapy can achieve excellent therapeutic effect through minimally invasive methods (such as percutaneous intervention). In fact, Mg powder would react with water to generate tiny amounts of $H_2$, which could be employed for the treatment of osteoarthritis and other diseases associated with inflammation[27–29]. However, the use of implantable metal Mg for antitumor gas therapy is challenging, due to the relatively slow reaction rate of Mg with water and thus insufficient $H_2$ generation.

In contrast to the above-mentioned electrolytic cells that require an external power supply, galvanic cells are a typical electrochemical etching method that can be used to generate $H_2$ gas spontaneously in the aqueous environment without an external power supply[30,31]. Herein, we design a micro-galvanic cell by in situ reduction of a small amount of platinum (Pt) on the surface of Mg rods (Fig. 1). The obtained Mg-based galvanic cell (MgG) can be etched by water to allow the effective generation of $H_2$ gas and $Mg(OH)_2$. After implanting MgG rods into tumors, the continuous generation of $H_2$ could inhibit mitochondrial respiration and impair redox homeostasis inside tumor cells, while the byproduct $Mg(OH)_2$ could neutralize the acidic tumor microenvironment (TME). In tumors with MgG rods implantation, an increase in the infiltration of $CD8^+$ T cells and a decrease in the number of immunosuppressive myeloid-derived suppressor cells (MDSCs) are observed, indicating the modulation of the immunosuppressive TME toward the immune-supportive

one, which is favorable for antitumor therapy. This strategy shows tumor inhibition effects in 4T1 and CT26 murine tumor models, and a malignant patient-derived xenograft (PDX) model of human origin. We also demonstrate that our $H_2$ therapy approach employing MgG rods could reduce the size of rabbit tumors. Last, the constructed MgG rods with biodegradable behavior show no appreciable toxicity. Our findings suggest that the widely used implanted Mg metal may be employed for successful and long-lasting $H_2$ therapy using a galvanic cell design.

## Results

**Preparation and characterization of MgG rods.** Galvanic cells generally consist of two kinds of metals with different electrode potentials[32,33]. To engineer MgG, Mg rods were placed in a neutral solution containing $PtCl_6^{2-}$ at room temperature, and then the Mg galvanic cells ($Mg^{2+}$/Mg: $-2.372$ V; $PtCl_6^{2-}$/$PtCl_4^{2-}$: 0.726 V; $PtCl_4^{2-}$/Pt: 0.758 V) were successfully constructed by in situ reducing Pt on the surface of Mg rods (Fig. 2a). It is known that the ion concentration ($PtCl_6^{2-}$) and the immersion time directly affect the Pt loading component via the self-assembly process, which may have an impact on the efficiency of $H_2$ generation. To optimize the MgG parameters, the $H_2$ generation performance of MgG rods prepared under different immersion time and concentrations of $PtCl_6^{2-}$ was determined using gas chromatography. The $H_2$ generation performance was enhanced with increasing ion concentrations and immersion time, and then reached a steady-state (Supplementary Figs. 1, 2). After MgG ($PtCl_6^{2-}$ concentration: 0.3%, immersion time: 1 min) construction, scanning electron microscopy (SEM) images and energy dispersive spectrometry (EDS) elemental mapping of a MgG rod showed homogeneous distribution of Mg and Pt elements (Fig. 2b and Supplementary Figs. 3, 4). Transmission electron microscopy (TEM) image and elemental mapping showed that the Pt nanoparticles (NPs, ~3 nm) were uniformly distributed over the Mg rods (Supplementary Fig. 5). XRD analysis also showed the peaks of Pt (JCPDS No. 04-0802) emerging from the as-prepared MgG rods, suggesting that Pt NPs were in situ reduced on the surface of the Mg rods (Fig. 2c and Supplementary Fig. 6).

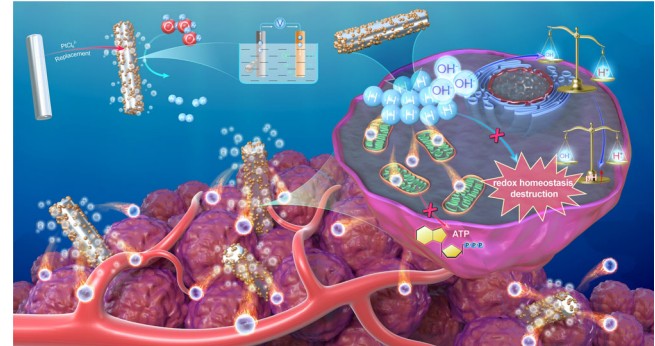

**Fig. 1 Schematic illustration to show the preparation of Mg galvanic cell (MgG) for tumor microenvironment modulation and enhanced cancer hydrogen therapy.** MgG is developed by decorating platinum on the surface of Mg rods, which allows the continuous generation of $H_2$ in an aqueous environment due to galvanic-cell-accelerated water etching of Mg. By implanting MgG rods into a tumor, $H_2$ molecules can be generated within the tumor, which induces mitochondrial dysfunction and intracellular redox homeostasis destruction. Meanwhile, the $Mg(OH)_2$ residue can neutralize the acidic tumor microenvironment (TME). Such MgG rods with the micro-galvanic cell structure enable hydrogen therapy to inhibit the growth of tumors.

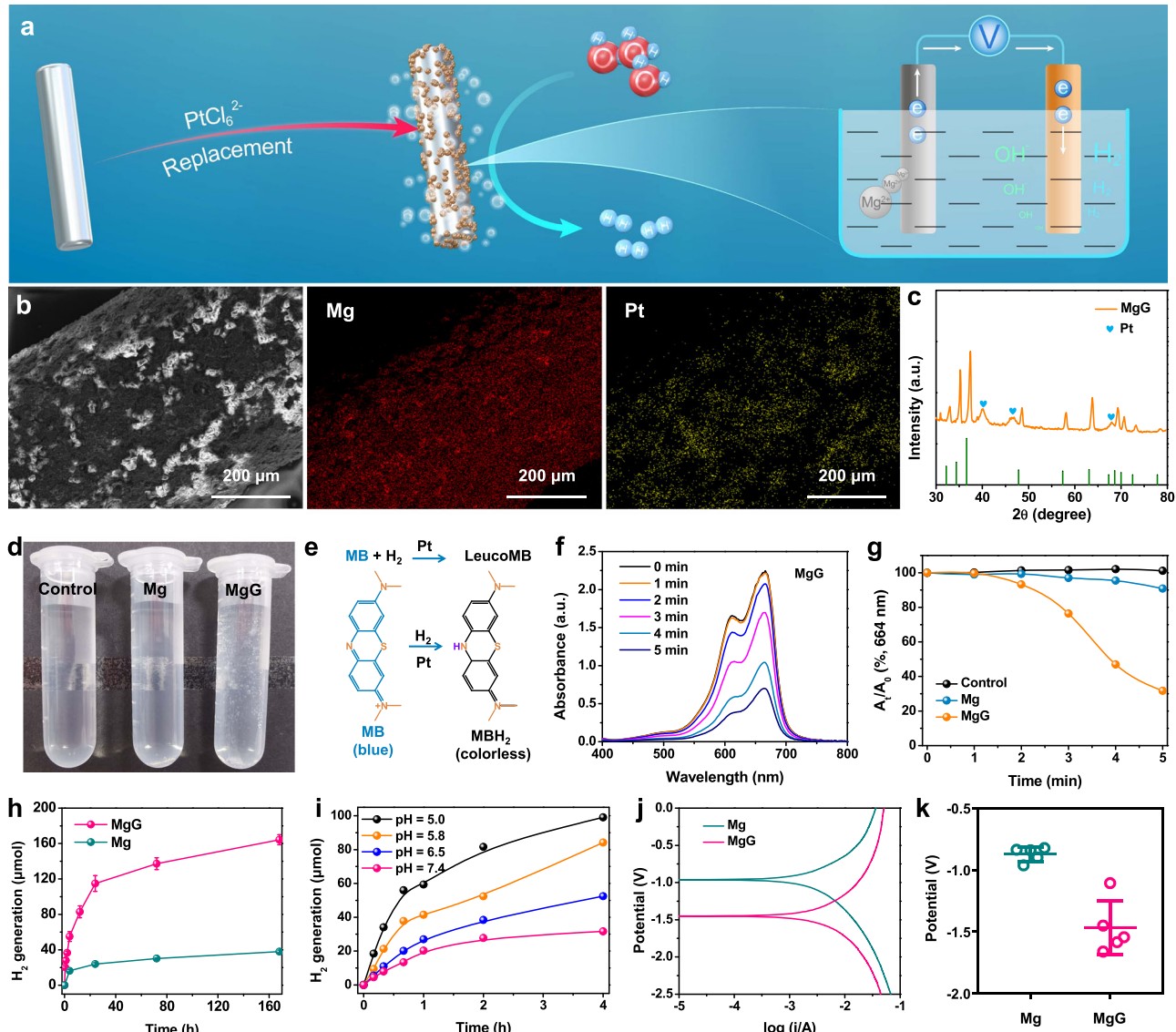

**Fig. 2 Preparation and characterization of MgG rods. a** Schematic illustration of the synthesis process of Mg galvanic cells and the $H_2$ production mechanism. **b** The representative SEM image and EDS element mapping of MgG rods from three independent samples. **c** The XRD patterns of MgG rods. **d** Photograph showing $H_2$ generation in PBS from Mg and MgG. **e** Schematic illustration of $H_2$ generation detected by the MB probe. **f** Time-dependent absorption spectra of the MB solution (pH = 6.5) with MgG rods added. **g** Comparison of MB reduction by Mg rods and MgG rods. **h** Time-dependent $H_2$ generation measured by gas chromatography from Mg or MgG rods in PBS (pH = 6.5, $n = 3$ biologically independent samples). **i** $H_2$ generation profiles of MgG rods in PBS solutions with different pH values measured by gas chromatography. **j** The Tafel curve of Mg rods and MgG rods. **k** The corrosion potentials of Mg rods and MgG rods ($n = 5$ biologically independent samples). Data are presented as mean values ± SD.

For Mg-Pt galvanic cells, once the metal electrode was placed into the aqueous environment, a spontaneous redox reaction occurred. Electrons ($e^-$) would flow out of the negative electrode (Mg electrode, $Mg-2e^- = Mg^{2+}$) and flow into the positive electrode (Pt electrode, $2H_2O + 2e^- = 2OH^- + H_2$), leading to the water etching of Mg and the generation of $H_2$. Indeed, by immersion in water, MgG rods could rapidly react with $H_2O$ and generate lots of $H_2$ gas bubbles, while bare Mg rods virtually generated no obvious bubbles (Fig. 2d). To further verify $H_2$ generation, methylene blue (MB) was used as the probe to detect $H_2$ produced from MgG. The blue-colored MB could be quickly reduced into colorless $MBH_2$ by $H_2$ (Fig. 2e)[34]. Notably, the MB characteristic peak in the MgG group was significantly reduced, while that in the bare Mg rod group only showed a slight decrease, indicating that MgG rods had a competent ability to generate $H_2$ gas by reacting with water (Fig. 2f, g and

Supplementary Fig. 7). Quantitative measurement of $H_2$ generation from the MgG and Mg rods was further conducted by gas chromatography. It was found that compared to bare Mg rods, MgG rods were able to generate more $H_2$ gas, continuously for more than 1 week (Fig. 2h). In addition, the $H_2$ production rate of MgG rods increased under lower pH, indicating that the MgG rods would show better $H_2$ generation capacity in the weak acid TME (Fig. 2i and Supplementary Fig. 8). During the $H_2$ generation process, no significant $O_2$ was generated because Mg was oxidized in the negative electrode reaction (Supplementary Fig. 9). In addition, the XPS spectra of MgG after the $H_2$ generation process showed strong signals from Mg (II), further proving that MgG rods were sacrificed during the generation of $H_2$ (Supplementary Fig. 10). Essentially, the galvanic cell is an electrochemical etching reaction, and a lower corrosion potential indicates an easier occurrence of oxidation corrosion

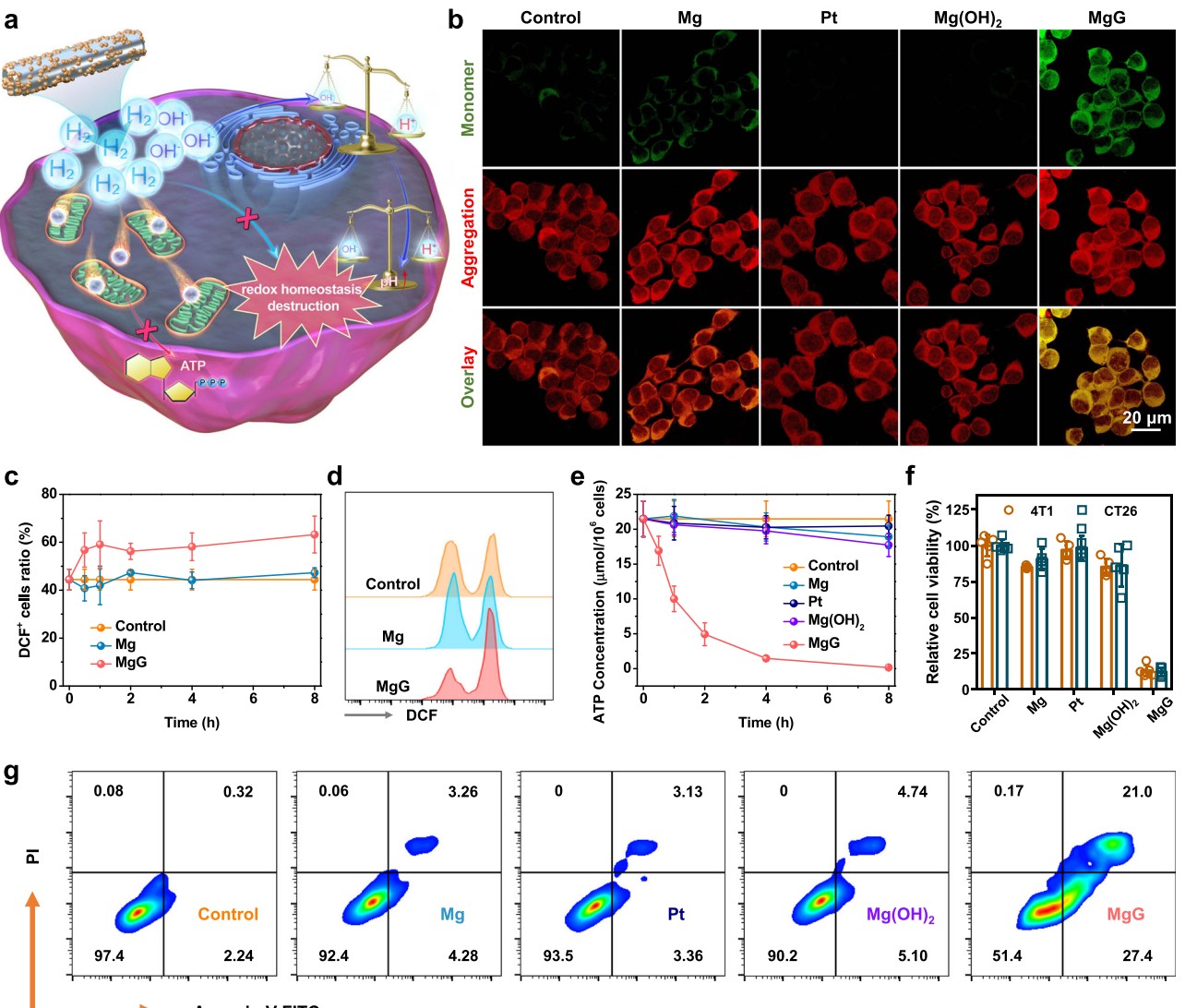

**Fig. 3 In vitro hydrogen therapy with MgG rods. a** Schematic illustration showing the mechanims of hydrogen therapy with MgG for cancer cell killing. **b** Detection of mitochondria membrane potentials by confocal fluorescence images of 4T1 cells stained with JC-1 dye. **c** Time-dependent changes of intracellular ROS in 4T1 cells during different treatments ($n = 5$ biologically independent samples). **d** Flow cytometry data to show DCF-positive 4T1 cells after different treatments. **e** Time-dependent changes of intracellular ATP contents in 4T1 cells during different treatments ($n = 5$ biologically independent samples). **f** Relative viabilities of 4T1 and CT26 cells after various treatments (control, Mg, Pt, Mg(OH)$_2$, and MgG, $n = 5$ biologically independent samples). **g** Flow cytometry analysis of 4T1 cells after various treatments using an Annexin V-FITC/PI kit. A representative image of three biologically independent samples from each group is shown in (**b**). Data are presented as mean values ± SD.

reaction[30,35]. Based on the Tafel curves, the corrosion potential of MgG was much lower than that of Mg, explaining the reason that MgG would react with water more efficiently (Fig. 2j, k).

**In vitro hydrogen therapy with MgG rods.** With continuous H$_2$ generation ability, we expected the utilization of MgG rods for further H$_2$ gas therapy (Fig. 3a). It has been reported that a high concentration of H$_2$ kills cancer cells by inhibiting cell mitochondrial respiration[16]. Therefore, the effect of MgG rods on inhibiting mitochondrial respiration was evaluated by measuring the membrane potential changes, with JC-1 dye as the fluorescent indicator. JC-1 dye forms red fluorescent aggregates in the normal mitochondrial membrane, while green fluorescent monomers exist in the damaged mitochondrial membrane[36]. Weak green fluorescence and strong red fluorescence were observed in the control group, indicating intact mitochondria of those cells

(Fig. 3b and Supplementary Fig. 11). While cells treated with Mg rods showed only a slight green fluorescence increase, the strongest green fluorescence was observed in cells after being treated with MgG rods, indicating that the continuous and abundant H$_2$ generation would significantly affect the cellular mitochondrial respiration. Moreover, rather weak green fluorescence occurred in the Pt and Mg(OH)$_2^-$ treated groups, demonstrating no significant mitochondrial damage caused by Pt or Mg(OH)$_2$ alone. On the other hand, while the equilibrium of redox homeostasis inside cells is important for their survival[37,38], it is known that breaking the intracellular redox homeostasis equilibrium would affect cell growth[39,40]. As expected, H$_2$ generated from MgG rods drastically altered the oxidative stress state of cancer cells with time (Fig. 3c, d and Supplementary Fig. 12). As a direct energy source for basic life activities, adenosine triphosphate (ATP) affects the living state of cancer cells. It was found that the ATP concentration within 4T1 cells showed a

remarkable decrease over time after being treated with MgG rods (Fig. 3e), suggesting that the cellular activity was inhibited by the continuously generated $H_2$, which thus caused a remarkable reduction in cellular energy. The above results demonstrated that the continuous $H_2$ generation by Mg-based micro-galvanic cells would cause effective mitochondrial dysfunction and intracellular redox homeostasis destruction.

To test their in vitro cell-killing efficacy, MgG rods were incubated with 4T1 murine breast cancer cells or CT26 colon adenocarcinoma cells, with bare Mg rods used as the control. There were only slight changes in cell viabilities after treatment with Mg rods, while most cells were damaged after being treated with MgG rods, indicating that the sustained $H_2$ generation at a high level could significantly lead to cell death (Fig. 3f). Notably, Pt NPs, $Mg(OH)_2$, and $Mg^{2+}$ exhibited only slight toxicity (Fig. 3f and Supplementary Fig. 13). Since alloying may increase the corrosion rate of Mg, the $H_2$ generation performance and cell-king effect of MgG rods and other commercialized Mg alloys (Mg, MgZnCa, MgAl) were evaluated. The MgG rods exhibited the strongest cell-killing effect, probably due to their superior $H_2$ generation capacity (Supplementary Figs. 14, 15). To further verify the ability of $H_2$ to kill cells, 4T1 and CT26 cells were treated with pure $H_2$ released from a hydrogen balloon. It was found that a high concentration of $H_2$ could indeed inhibit the proliferation of cancer cells (Supplementary Fig. 16). Moreover, the flow cytometry analysis further confirmed that MgG treatment caused significant apoptosis of 4T1 and CT26 cells (Fig. 3g and Supplementary Fig. 17). All of the above results demonstrated that the continuous $H_2$ generation originating from MgG rods could lead to the effective killing of cancer cells by inhibiting mitochondrial respiration and disrupting redox homeostasis.

**In vivo hydrogen therapy with MgG rods**. Next, we wondered whether such MgG rods could be used for in vivo antitumor therapy. First, ultrasonic imaging was performed to monitor hydrogen gas generation in the tumor, as gas bubbles would offer strong contrast under ultrasound imaging. After MgG rods were implanted into the tumor, strong ultrasound signals appeared in the tumor for more than 48 h, suggesting efficient gas production (Fig. 4a, b and Supplementary Fig. 18). In contrast, the ultrasound signals in Mg rod implanted tumors were much weaker, indicating the insignificant $H_2$ generation from the bare Mg rods (Fig. 4b and Supplementary Fig. 19). In addition, SEM images and EDS element mapping of MgG rods after implantation into the tumor for 4 and 24 h showed that the galvanic cell structure accelerated the corrosion of Mg to generate enough $H_2$. Despite the obvious corrosion of MgG rods, Pt NPs remained on the surface of those rods at 24 h, allowing further in vivo corrosion of MgG and thus the generation of $H_2$ for gas therapy (Supplementary Fig. 20). Notably, the MgG rods implanted into the tumor indeed showed efficient $H_2$ generation at 4 and 24 h as evidenced by gas chromatography (Supplementary Fig. 21). After etching MgG, we expected that the residual product $Mg(OH)_2$ as an alkaline substance would be able to neutralize the acidic tumor pH. Therefore, the time-dependent pH change curves of tumors in mice post intratumoral implantation with MgG rods were monitored by a pH microsensor (Fig. 4c). The tumor pH was rapidly neutralized to the neutral level and could be maintained for up to 72 h. Furthermore, ex vivo fluorescence imaging of tumors after MgG rods implantation was performed using a pH-responsive fluorescent probe (Fig. 4d, e). Consistently, the tumor pH reached the peak level at 24 h post-MgG implantation, and such neutralization effect could be maintained for 72 h. Notably, although the metal-water reaction is exothermic, there was no

temperature change at the tumor site after intratumoral implantation of MgG rods, suggesting that the reaction of MgG with water was moderate and would not induce a significant thermal effect (Supplementary Fig. 22). Solid tumors are characterized by an acidic microenvironment, which may impede effective antitumor T-cell immune responses. Specifically, $CD8^+$ T cells tend to become anergic and apoptotic when exposed to a low pH environment. The neutralization of tumor acidity would be favorable for antitumor immune responses[41,42]. Therefore, MgG-induced antitumor immune responses were evaluated on 4T1 tumor-bearing mice. These mice post-implantation of MgG at day 0 were sacrificed at day 6, and their tumors were collected to determine the levels of immune cells. Compared with the other groups, the $Mg(OH)_2$ group showed a slight regulatory effect, while the tumors from the MgG group exhibited reduced populations of MDSCs and an increased percentage of total T cells (especially $CD8^+$ T cells). MDSCs, as immunosuppressive cells, could promote tumor progression by inhibiting antitumor immunity, while T cells (especially $CD8^+$ T cells) would ensure effective tumor cell killing. Therefore, MgG implantation would thus modulate the immunosuppressive TME for enhanced tumor therapy (Fig. 4f–i, and Supplementary Figs. 23, 24).

Next, we carefully investigated the antitumor efficacy of $H_2$ gas therapy based on the micro-galvanic cell strategy. Mice bearing subcutaneous 4T1 tumors were randomly divided into seven groups ($n = 10$) when their tumor volumes reached ~150 mm³: (I) Control; (II) Mg rods implantation; (III) Pt NPs (remaining after dissolving MgG rods); (IV) $Mg(OH)_2$ (remaining after the MgG rods produced $H_2$); (V) MgG rods implantation; (VI) Mg rods implantation (three times, 0th day, 4th day, and 8th day); and (VII) MgG rods implantation (three times, 0th day, 4th day, and 8th day). For Mg or MgG rods implantation, two Mg or MgG rods ($D = 0.5$ mm, $L = 4$ mm) were implanted into each tumor (~7 mm × 7 mm) (Supplementary Fig. 18). Pt NPs- and Mg-implanted groups had no significant tumor-suppressing effect (Fig. 4j, k). The tumors of the $Mg(OH)_2$ group showed slightly delayed growth, likely due to the neutralization of tumor pH. Remarkably, the tumor growth of the MgG rods implantation group was significantly inhibited, especially for those with three times of MgG rod implantations. Notably, MgG implantation could significantly prolong animal survival, and four out of ten mice survived for more than 60 days after three times of MgG-implantations (Fig. 4k and Supplementary Fig. 26). Hematoxylin and eosin (H&E) staining and TdT-mediated dUTP nick-end labeling (TUNEL) staining of tumor slices further confirmed that MgG rods implantation could induce severe tumor cell apoptosis (Fig. 4l and Supplementary Fig. 25). In addition, there was no significant body weight loss for mice implanted with MgG rods, indicating no obvious side effects induced by MgG-based $H_2$ therapy (Supplementary Fig. 27). Moreover, mice bearing subcutaneous 4T1 tumors expressing firefly luciferase (Luc-4T1) were also used to intuitively display the therapeutic efficacy. MgG rods implantation group showed the weakest bioluminescence signal, and demonstrated the best tumor growth inhibition (Fig. 4m).

**In vivo hydrogen therapy of different tumor models with MgG rods**. Encouraged by the superior therapeutic performance of MgG rods for the treatment of 4T1 tumors, we then evaluated the therapeutic efficacy of MgG-induced $H_2$ therapy on other tumor models. CT26 tumor-bearing mice were randomly assigned into five groups ($n = 10$) when their tumor volume reached ~150 mm³ (Fig. 5a). Following a similar trend to the results of the 4T1 model, Mg rods implantation was essentially unable to inhibit tumor growth, even after three times of implantation. As

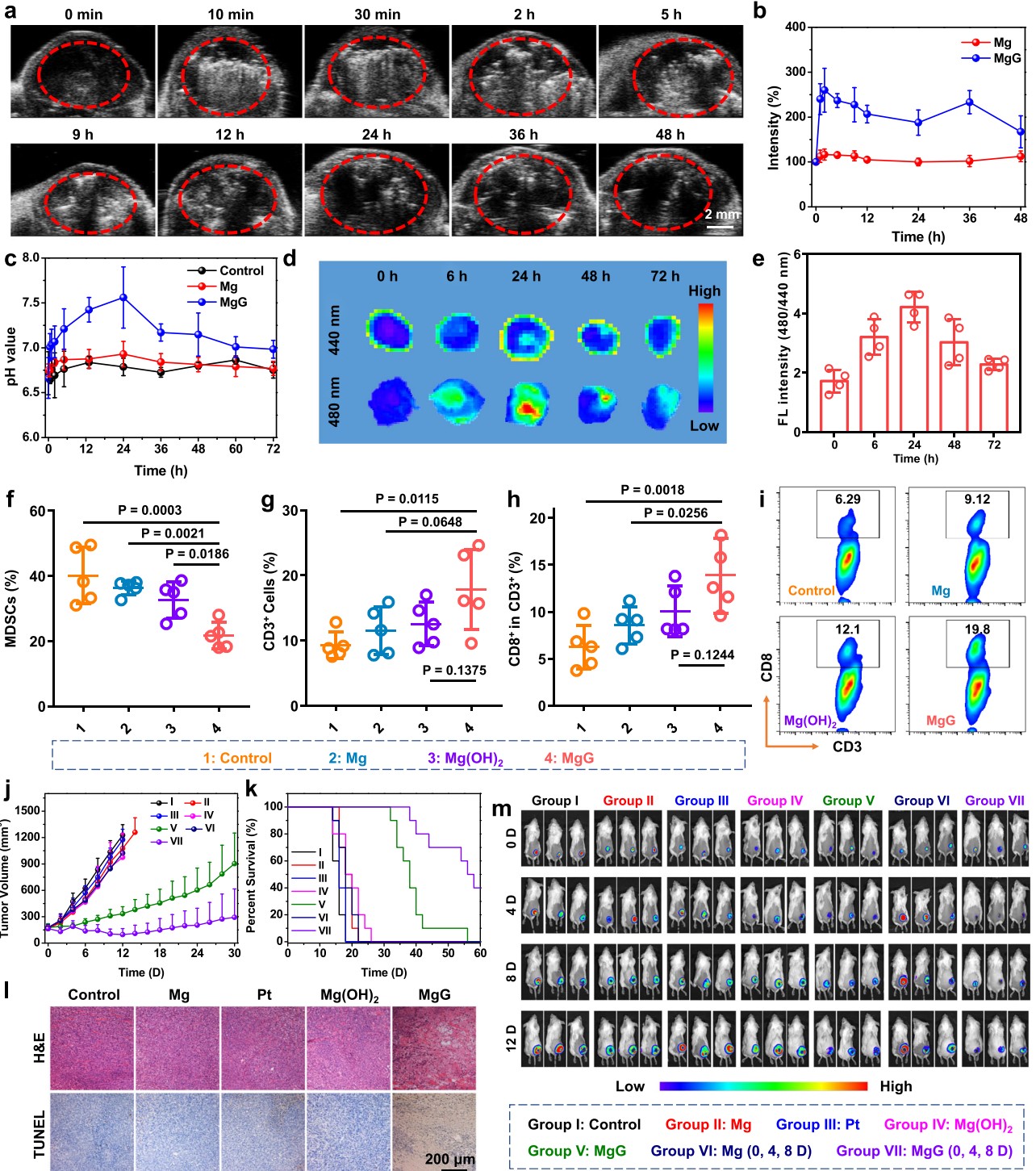

**Fig. 4 In vivo hydrogen therapy with MgG rods. a** In vivo time-dependent ultrasonic imaging of 4T1 tumor-bearing mice after intratumoral implantation with MgG rods. **b** Quantitative analysis of signal intensities based on ultrasonic imaging data. **c** Time-dependent pH value changes of tumors in mice post intratumoral implantation with Mg rods or MgG rods. **d** Ex vivo fluorescence images of tumors after implantation of MgG for different periods of time. **e** Tumor fluorescence (FL) signal ratios (480/440) based on ex vivo fluorescence imaging data in (C). **f**–**h** The quantification results of MDSCs (CD45$^+$CD11b$^+$Gr-1$^+$, **f**), T cells (CD3$^+$, **g**), CD8$^+$ T cells (CD3$^+$CD8$^+$, **h**) by flow cytometry on day 6 post-MgG implantation. **i** The flow cytometric analysis results of CD8$^+$ T cells (CD3$^+$CD8$^+$) within the tumors after different treatments. **j** The growth curves of tumors after various treatments. **k** Survival rates of tumor-bearing mice after various treatments. **l** Microscopy images of H&E and TUNEL stained tumor slices collected from mice post different treatment groups. **m** In vivo bioluminescence images of mice bearing subcutaneous 4T1 tumors expressing firefly luciferase (Luc-4T1) to display the therapeutic efficacy of mice after various treatments. $n = 3$ biologically independent animals in (**b**, **c**). $n = 4$ biologically independent animals in (**e**). $n = 5$ biologically independent animals in (**f**–**h**). $n = 10$ biologically independent animals in (**j**). A representative image of three biologically independent animals from each group is shown in (**a**, **l**). Data are presented as mean values ± SD. $P$ values calculated by the two-tailed student's $t$ test are indicated in the Figures.

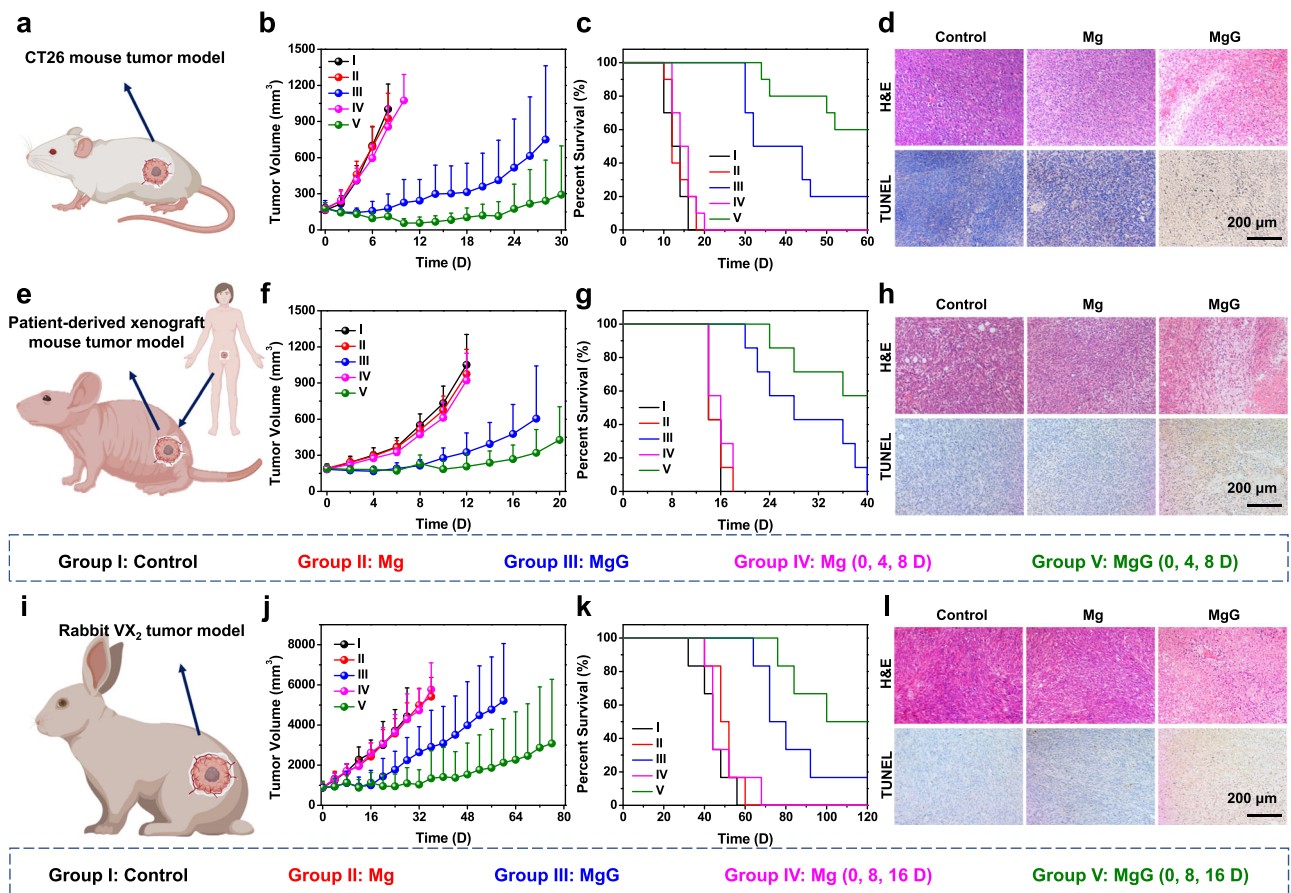

**Fig. 5 In vivo hydrogen therapy of different tumor models with MgG rods. a** Scheme of the subcutaneous CT26 mouse tumor model for MgG-based hydrogen therapy. **b**, **c** Tumor growth curves (**b**) and survival rates (**c**) of CT26 tumor-bearing mice after different treatments as indicated. **d** Microscopy images of H&E and TUNEL stained CT26 tumor slices collected from different groups. **e** Scheme of the PDX tumor model in mice for MgG-based hydrogen therapy. **f**, **g** Tumor growth curves (**f**) and survival rates (**g**) of nude mice after different treatments. **h** Microscopy images of H&E and TUNEL stained PDX tumor slices collected from different groups. **i** Scheme of the VX$_2$ liver rabbit tumor model for MgG-based hydrogen therapy. **j**, **k** Tumor growth curves (**j**) and survival rates (**k**) of rabbits after different treatments. **l** Microscopy images of H&E and TUNEL stained VX$_2$ tumor slices collected from different groups. $n = 10$ biologically independent animals in (**b**). $n = 7$ biologically independent animals in (**f**). $n = 6$ biologically independent animals in (**j**). A representative image of three biologically independent animals from each group is shown in (**d**, **h**, **l**). Data are presented as mean values ± SD.

expected, MgG rods implantation significantly inhibited the tumor growth, with ~40% of tumors being completely eliminated after three times of MgG implantations (Fig. 5b, c and Supplementary Figs. 28, 29). Tumor cells were severely damaged by MgG rods implantation while the other groups showed no obvious cell damage as determined by H&E and TUNEL staining (Fig. 5d and Supplementary Fig. 25). In addition, there was no significant body weight loss for the mice implanted with MgG rods, indicating no obvious side effects induced by MgG-based H$_2$ therapy (Supplementary Fig. 30). Therefore, the MgG-induced H$_2$ therapy strategy on CT26 model also showed great therapeutic effects.

Next, we further tested our MgG-induced H$_2$ therapy strategy to treat a more clinically relevant tumor model. A cervical PDX tumor model[43,44], in which tumors from real human patients are implanted into immunodeficient mice, was further used to evaluate the efficacy of MgG-induced H$_2$ therapy. To establish a PDX tumor model, tumor tissues were surgically resected from a cervical cancer patient and cut into pieces, which were implanted into nude mice (Fig. 5e). For MgG-induced H$_2$ therapy, PDX tumor-bearing nude mice were randomly assigned into five groups ($n = 7$) when the tumor volume reached ~150 mm$^3$, and treated via the same parameters as that used for the treatment of the CT26 tumor model. Although it was difficult to completely

eliminate those PDX tumors, MgG rod implantation still significantly extended the survival time of mice compared with that of the other control groups (Fig. 5f, g and Supplementary Fig. 31), and caused obvious tumor damage as revealed by H&E and TUNEL staining of tumor slices (Fig. 5h and Supplementary Fig. 25). In addition, there was no significant decrease of body weight for the mice implanted with MgG rods, indicating good safety of MgG-based H$_2$ therapy (Supplementary Fig. 32).

In addition to the mouse model experiment, the therapeutic efficacy of our strategy in the treatment of a larger animal tumor model was also evaluated. Rabbit-bearing VX$_2$ tumors were randomly allocated and treated by intratumoral implantation of MgG rods (Fig. 5i). Due to the large rabbit tumor sizes (~800 mm$^3$, five times larger than that on mice), three Mg or MgG rods (L = 8.0 mm, D = 0.8 mm) were implanted into each tumor (~12 mm × 12 mm) for rabbit treatment (Supplementary Fig. 18). MgG-implanted tumors were effectively inhibited by continuous H$_2$, while tumors of the control group or Mg rods implantation group showed rapid growth (Fig. 5j, k and Supplementary Fig. 33). In fact, tumors on two out of six rabbits were completely eliminated without re-growth and survived for over 120 days after H$_2$ treatment with three implantation cycles (Fig. 5k). Moreover, the tumor slices collected from rabbits treated with MgG rods also showed severe histological damage and a high level of cell apoptosis compared to the

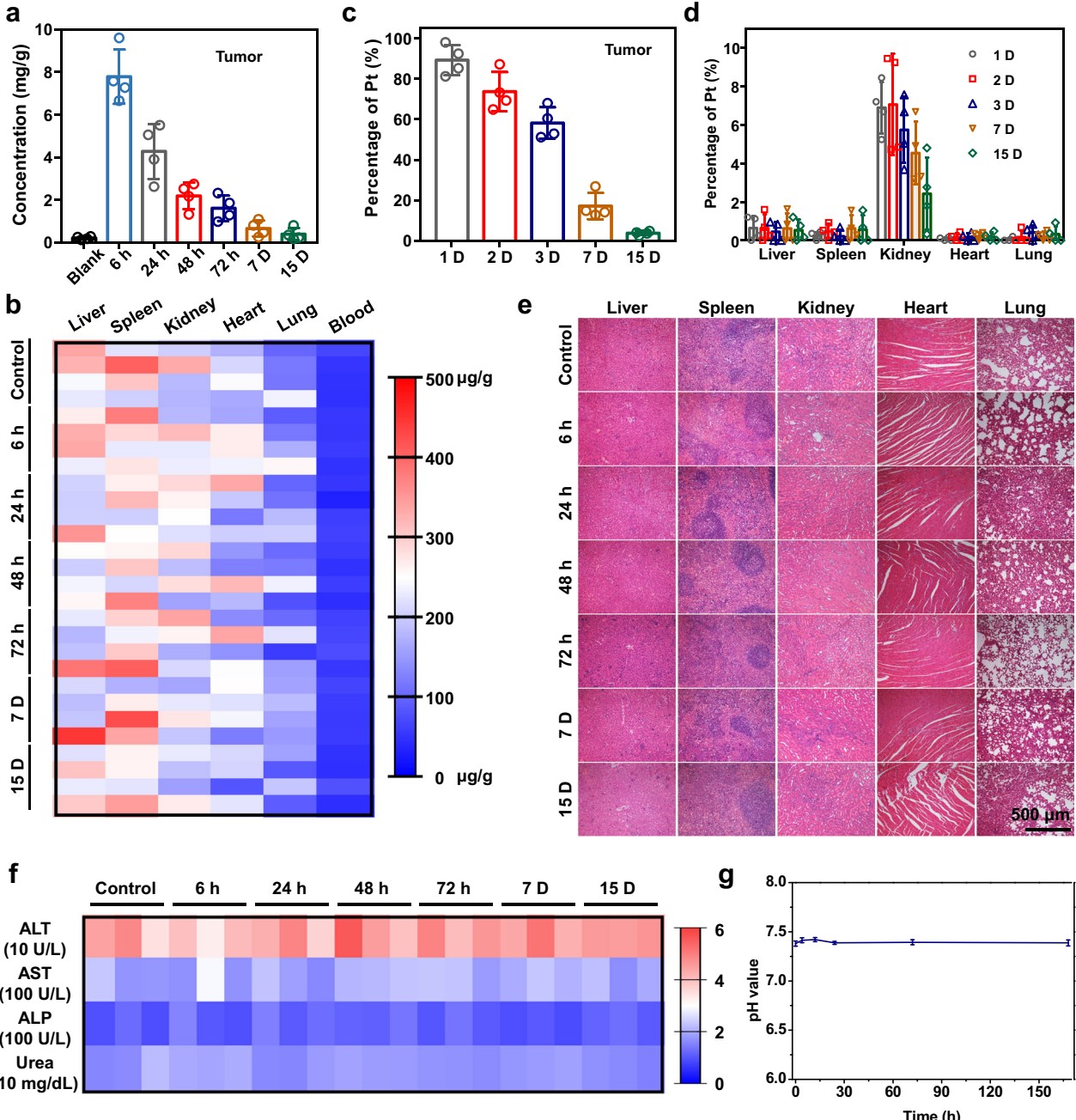

**Fig. 6 In vivo safety and biodegradation behaviors of MgG rods. a** The Mg levels in tumors of mice after being implanted with MgG rods for different periods of time. **b** The Mg levels in major organs and blood of mice after being implanted with MgG rods for different periods of time. The Mg contents were measured by ICP-OES. No notable variation of Mg levels was found in the blood or organs of implanted mice compared to the control. **c** The Pt levels in tumors of mice after being implanted with MgG rods for different periods of time (1, 2, 3, 7, and 15 D). **d** The Pt levels in major organs of mice after being implanted with MgG rods for different periods of time (1, 2, 3, 7, and 15 D). **e** Hematoxylin and eosin (H&E) staining of mouse major organs (liver, spleen, kidney, heart, and lung) to examine the histological changes after implantation of MgG rods in mice. **f** Blood biochemistry data of Balb/c mice after being implanted with MgG rods for different periods of time (0, 6 h, 24 h, 48 h, 72 h, 7 days, and 15 days). The measured indexes included alanine aminotransferase (ALT), aspartate aminotransferase (AST), alkaline phosphatase (ALP), and blood urea. **g** Blood pH value variation of Balb/c mice after being implanted with MgG rods for different periods of time (0, 4 h, 12 h, 24 h, 72 h, and 7 days). $n = 4$ biologically independent animals in (**a**, **c**–**d**). $n = 3$ biologically independent animals in (**g**). A representative image of three biologically independent animals from each group is shown in (**e**). Data are presented as mean values ± SD.

other treatment groups (Fig. 5l and Supplementary Fig. 25). All these results showed that VX₂ tumors with larger sizes could also be effectively inhibited after MgG rods implantation, illustrating the capability of MgG-based H₂ therapy to kill large-sized tumors.

**In vivo safety and biodegradation behaviors of MgG rods.** Last but not the least, it is important to evaluate the safety of MgG rods before future clinical trials. 4T1 tumor-bearing mice were intratumorally implanted with MgG rods for different time

points. First, the Mg levels in the tumors of mice after being implanted with MgG rods for different times were measured by inductively coupled plasma optical emission spectrometry (ICP-OES) (Fig. 6a). The Mg content in the tumor gradually decreased over time, and most of the MgG rods were degraded within 15 days. Such degradable behavior of MgG rods was mainly due to the galvanic-cell-accelerated water etching of Mg after implantation. The formed $Mg(OH)_2$ would further react with the $H^+$ in the TME to form $Mg^{2+}$ ions. Moreover, there was no significant increase of Mg levels in the blood and main organs (liver, spleen, kidney, heart, and lung) after MgG rods were implanted into the body, indicating that $Mg^{2+}$ generated from the degraded MgG would not affect the equilibrium of Mg levels in the body (Fig. 6b). In addition, the biodistribution of Pt NPs after degradation from MgG in the tumor and the main organs was studied. Pt content in the tumor was gradually decreased over time, and the major distribution of Pt was detected in the kidney, indicating the renal clearance of those Pt NPs with their ultra-small sizes (Fig. 6c, d). Meanwhile, no obvious histological damage was observed on organ slices from the MgG rods treated mice (Fig. 6e). Other main indicators were further evaluated as to whether rapid degradation would result in potentially toxic side effects. Alanine aminotransferase (ALT), aspartic aminotransfer-ase (AST), and alkaline phosphatase (ALP), all of the important liver function markers, were within the reference range com-parable to the control group, indicating that MgG rods implanted caused no significant hepatotoxicity (Fig. 6f). As an indicator of kidney function, the blood urea levels in the blood of treated mice were also within the normal range (Fig. 6f). For the hematological assessment, the white blood cells, red blood cells, hemoglobin, hematocrit, mean corpuscular volume, mean corpuscular hemo-globin (MCH), mean corpuscular hemoglobin concentration, and platelets were selected (Supplementary Fig. 34), and all these hematological assay data were found to be normal in the MgG-treated groups compared with that in the control group. More-over, there was no obvious pH value variation in blood of Balb/c mice after being implanted with MgG rods (Fig. 6g). Taken together, all the above results illustrated that MgG rods could be rapidly degraded after implantation and displayed no obvious toxicity to the treated animals.

## Discussion

Mg and its alloys have been extensively used as implantable metals in clinic for over 100 years[22,23,45,46]. As one of the essential elements, Mg participates in almost all metabolic processes[47]. In our study, the degradation products of MgG did not show any long-term hazardous effect to the body. Importantly, our strategy based on MgG could achieve a therapeutic effect without the need to introduce external stimulations nor combine with other treatments, and may be more convenient than current gas-induced therapies. Local administration of the implantable $H_2$ therapy can achieve a therapeutic effect through minimally invasive methods (such as percutaneous intervention). However, the implantable $H_2$ therapy still has some disadvantages, such as the difficulty of operation and intervention for tumors in the important organs or complex parts of the body (such as the mediastinum, pancreas, and intracranial tumors). In addition, cancer immunotherapy has presented tremendous promises by boosting the immune system to fight cancer in recent years[48–51], and the acidic tumor pH is an immunosuppressive feature of solid tumors. The strategy of neutralizing tumor acidity with MgG rods would thus be favorable for the antitumor immune response[41,52]. Above all, our work proposed that the galvanic cell based on implant metal Mg could be a feasible approach to achieving safe and effective $H_2$ therapy, which may be combined with other therapeutic strategies such as cancer immunotherapy in future clinical trials.

In summary, we provided an $H_2$ gas therapy technique based on Mg micro-galvanic cells. The development of numerous micro-galvanic cells after in situ reduction of Pt on the surface of Mg rods would diminish the corrosion potential of Mg, con-siderably boosting the reaction rate of Mg with $H_2O$ without the use of external auxiliary equipment. After implanting MgG rods into tumors, the sustained generation of $H_2$ could result in the inhibition of cancer cell respiration by significantly lowering the mitochondrial membrane potential, affecting ATP synthesis, and disrupting the balance of redox homeostasis within tumor cells, while the residual product $Mg(OH)_2$ could neutralize the acidic tumor pH. The increased tumor infiltration of $CD8^+$ T cells and the decreased number of MDSCs in MgG-implanted tumors suggest the modulation of the immunosuppressive TME by such treatment. The MgG-induced $H_2$ therapeutic strategy with con-tinuous and convenient $H_2$ generation was then employed for in vivo tumor ablation, not only for inhibiting 4T1 and CT26 tumors, but also for treating the human-derived malignant PDX model. In addition to killing tumors on mice, $VX_2$ tumors of bigger sizes growing in rabbits could also be effectively reduced after the implantation of MgG rods.

## Methods

**Materials**. Magnesium (Mg) rods were obtained from JingJun Materials Tech-nology Co. Ltd. (Suzhou, China). Chloroplatinic acid ($H_2PtCl_6$) and 2′,7′-dichlorofluorescein diacetate (DCFH-DA) were purchased from Sigma-Aldrich Co., Ltd. Sodium hydroxide (NaOH) was purchased from Sinopharm Chemical Reagent Co., Ltd. Magnesium chloride ($MgCl_2$), magnesium hydrate [$Mg(OH)_2$], and MB were purchased from Aladdin Co., Ltd. All chemicals and reagents were of analytical grade and used without further purification. Anti-CD3-FITC (Biolegend, clone 17A2, Catalog: 100204), anti-CD4-APC (Biolegend, clone GK1.5, Catalog: 100412), anti-CD8-PE (Biolegend, clone 53–6.7, Catalog: 100708), anti-CD45-FITC (Biolegend, clone I3/2.3, Catalog: 147710), anti-CD11b-PE (Biolegend, clone M1/70, Catalog: 101208), anti-Gr-1-APC (Biolegend, clone RB6-8C5, Catalog: 108412) were obtained from Biolegend and diluted at 1:300 for cell staining.

**Engineering of Mg galvanic cells (MgG)**. In the experiments, the pH of a $H_2PtCl_6$ solution was adjusted to neutral by NaOH. Then Mg rods were put into the neutral solution containing $PtCl_6^{2-}$. To optimize the MgG parameters, the $H_2$ generation performance of MgG rods prepared under different immersion time (0.2, 0.5, 1.0, 3.0, and 5.0 min) and $PtCl_6^{2-}$ ion concentration (0.01, 0.03, 0.1, 0.3, and 1.0%) conditions were performed using gas chromatography.

**Characterization**. Scanning electron microscope (SEM) imaging and elemental mapping were carried out by Zeiss Supra 55. Transmission electron microscope (TEM) image and elemental mapping were carried out by TALOS 200X. The Chemical composition and phase analysis were examined by using a PANalytical X-ray diffractometer (XRD). UV-vis-NIR absorbance spectra were detected by PerkinElmer Lambda 750 UV-vis-NIR spectrophotometer. X-ray photoelectron spectroscopy (XPS) was detected by Thermo Scientific K-Alpha. The absolute Mg and Pt contents were determined by ICP-OES. The temperature and thermal images were detected and recorded by an IR thermal camera (Fotric 225).

**Measurement of $H_2$ generation**. Qualitative measurement of the $H_2$ generation from the MgG rods in PBS was conducted using the MB probe. Blue MB could be quickly reduced into colorless $MBH_2$ by $H_2$ under the catalysis of Pt nanocrystals. The UV-vis-NIR spectra of the samples were scanned to monitor the decreasing trend at 664 nm, which indicated $H_2$ generation. In the meanwhile, quantitative measurement of $H_2$ generation from the Mg rod or MgG rods in PBS was mea-sured by gas chromatography (GC 2060, TCD detector, and Ar carrier).

**Measurement of Tafel curves**. The Tafel curves of Mg rods and MgG rods were recorded using a CHI 660E electrochemical analyzer in a three-electrode system. Mg rods or MgG rods were used as working electrodes, with the Ag/AgCl/KCl (3.5 M) electrode working as a reference electrode and Pt foil working as a counter electrode.

**In vitro $H_2$ gas therapy**. 4T1 murine breast cancer cells (SCSP-5056) and CT26 colon cancer cells (TCM37) were obtained from the Cell Bank, Shanghai Institutes for Biological Sciences, Chinese Academy of Sciences. Luciferase-transfected 4T1 cells (Luc-4T1) was obtained from PerkinElmer Co. as a gift, Rabbit $VX_2$ liver

cancer cells (MZ-0769) was obtained from Ningbo Mingzhou Biological Technology Co., Ltd. All the cell lines were authenticated and checked for mycoplasma contamination by their suppliers. No mycoplasma contamination was found before use.

To demonstrate its cytotoxicity by the indication of cell viability, 4T1 cells or CT26 cells were incubated with $MgCl_2$ at different concentrations for 24 h. For in vitro $H_2$ gas therapy, 4T1 cells and CT26 cells were incubated with Mg rods, Pt, $Mg(OH)_2$, or MgG rods, for 24 h. The relative cell viabilities were determined by the standard methyl thiazolyl tetrazolium assay. For the detection of mitochondria membrane potential, 4T1 cells after various treatments for 8 h were stained with JC-1 (monomer: green, aggregation: red). All fluorescent images were acquired by Confocal Laser Scanning Microscope (CLSM, Zeiss Axio-Imager LSM-800). In addition, the intracellular ATP levels were also detected according to the ATP content detection kit (Solarbio). For ROS detection, the treated 4T1 cells and CT26 cells were stained with DCFH-DA (20 µM) for 30 min, and further analyzed by flow cytometry. Meanwhile, flow cytometry analysis of 4T1 cells and CT26 cells after various treatments were used to quantitatively evaluate the number of the dead cells.

**Animal experiments**. Female Balb/c mice (6–8 weeks), female Balb/c nude mice (6–8 weeks) were purchased from Changzhou Cavins Biological Technology Co. Ltd, and New Zealand white rabbits (3–5 months) were purchased from Suzhou Zhenhu Laboratory Animal Science and Technology Co. Ltd. All animal experiments were carried out following protocols approved by Laboratory Animal Center of Soochow University (No. ECSU-2020000175). Mice were housed in groups of 5 mice per individually ventilated cage in a 12 h light-dark cycle (8:00–20:00 light; 20:00–8:00 dark), with constant room temperature (21 ± 1 °C) and relative humidity (40–70%). All mice had access to food and water ad libitum. In our experiment, the maximum tumor burden was 1500 mm³ in mice and 8000 mm³ in rabbits, both lower than the maximal tumor burden permitted by Laboratory Animal Center of Soochow University. In some cases, this limit has been exceeded the last day of measurement and the mice were immediately euthanized.

To construct the subcutaneous 4T1 breast tumor model, 4T1 cells or Luc-4T1 cells ($2 \times 10^6$) dispersed in ~50 µL of PBS were subcutaneously inoculated to the back of each female balb/c mouse. To construct the subcutaneous CT26 colon adenocarcinoma model, CT26 cells ($3 \times 10^6$) dispersed in ~50 µL of PBS were subcutaneously inoculated to the back of each female balb/c mouse. To develop the rabbit tumor model, a fish-like mass of tumor tissue (~4 mm × 4 mm × 4 mm) was cut into fragments dispersed in ~200 µL of PBS and then injected into the legs of each New Zealand white rabbit. All animals were randomized before experiments. All experiments were conducted in compliance with all relevant ethical regulations.

**Patient-derived xenograft (PDX)**. Patients' cervical tumor tissue was acquired from patients in the First Affiliated Hospital of Soochow University after obtaining informed consent. Patients' cervical tumor tissue was used following protocols approved by Ethics Committee of the First Affiliated Hospital of Soochow University (No. HESU-2020001032) and good clinical practice approved by the China Food and Drug Administration. For the PDX tumor model, Fresh tumor tissue (after surgical resection) was kept in the growth medium and cut into pieces (~2 mm × 2 mm × 2 mm) in a sterile dish. The tumor pieces were then implanted to the back of each female nude mouse under general anesthesia using forceps. The skin incision was closed using an absorbable suture.

**Ultrasonic imaging**. For in vivo ultrasonic imaging, 4T1 tumor-bearing mice were imaged under the Vevo LAZR small animal ultrasonic imaging system after implantation with MgG rods (D = 0.5 mm, L = 4.0 mm), with two rods for each tumor.

**Quantitative measurement of tumor pH**. The pH of 4T1 tumor-bearing mice was measured by a pH microsensor (PreSens, Germany) after implantation with MgG rods (D = 0.5 mm, L = 4.0 mm), with two rods for each tumor.

**Qualitative measurement of tumor pH**. For qualitative measurement of tumor pH, 4T1 tumor-bearing mice were implanted with MgG rods (D = 0.5 mm, L = 4.0 mm) for different periods of time, with two rods for each tumor. A fluorescent probe was injected into the tumor half an hour in advance. Fluorescent imaging of the tumor was performed under the Lumina III in vivo optical imaging system.

**In vivo $H_2$ gas therapy**. For in vivo $H_2$ therapy, 4T1 tumor-bearing mice were randomly divided into seven groups (ten mice per group) when the tumor volume reached ~150 mm³: (I) Control; (II) Mg rods implantation; (III) Pt NPs (remained after dissolving MgG rods); (IV) $Mg(OH)_2$ (remained after the MgG rods produced $H_2$); (V) MgG rods implantation; (VI) Mg rods implantation (three times, 0th day, 4th day, and 8th day); (VII) MgG rods implantation (three times, 0th day, 4th day, and 8th day). For CT26-tumor model and PDX model, mice were randomly divided into five groups (CT26-tumor model: ten mice per group, PDX model: seven mice per group) when the tumor volume reached ~150 mm³, respectively: (I)

Control; (II) Mg rods implantation; (III) MgG rods implantation; (IV) Mg rods implantation (three times, 0th day, 4th day, and 8th day); (V) MgG rods implantation (three times, 0th day, 4th day, and 8th day). The tumor sizes and body weights were measured every 2 days. The tumor volumes were calculated by the formula: Volume = length × width²/2. The mouse was sacrificed when its tumor size exceeded 1500 mm³.

For the rabbit tumor model, $VX_2$ tumor-bearing New Zealand white rabbits were randomly divided into five groups (six rabbits per group) when the tumor volume reached ~800 mm³, respectively: (I) Control; (II) Mg rods implantation; (III) MgG rods implantation; (IV) Mg rods implantation (three times, 0th day, 8th day, and 16th day); (V) MgG rods implantation (three times, 0th day, 4th day, and 8th day). The tumor sizes and body weights were measured every 4 days. The tumor volumes were calculated by the formula: Volume = Length × Width²/2. The rabbit was sacrificed when its tumor size exceeded 8000 mm³. The mice were implanted with two Mg rods or MgG rods (D = 0.5 mm, L = 4 mm) for each tumor, and the rabbits were implanted with three MgG rods (D = 0.8 mm, L = 8 mm) for each tumor by a simple implantation approach (Fig. S18). For the H&E and TUNEL staining, the formalin-fixed tumors were embedded in paraffin blocks, sectioned into slices, stained following the standard protocol, and visualized by fluorescence optical microscope (DM4000M, Laica, Germany).

**Biodegradability and biocompatibility study**. 4T1 tumor-bearing mice were implanted with MgG rods (two rods per mouse, D = 0.5 mm, L = 4.0 mm) intratumorally. Major organs including liver, spleen, heart, lung, and kidney of each group were dissected at different time points and cut into two halves. Partial of the organs were fixed in 4% neutral buffered formalin, embedded into paraffin, sectioned by routine procedures for further H&E staining, and finally observed by fluorescence optical microscope (DM4000M, Laica, Germany). The other half of the organs and tissues were weighed, solubilized in aqua regia and measured by ICP-OES to determine Mg levels in those organs. The tumor and blood were also collected and weighed, solubilized in aqua regia, and measured by ICP-OES to determine Mg and Pt levels. Meanwhile, the blood samples were collected for blood biochemistry tests, which were conducted by Servicebio Biotechnology Co., Ltd. (Wuhan).

**Software**. All statistical analyses were performed on Origin 8.5, Excel 2019, Image J (version 6) and GraphPad Prism 7. Chemical structures were drawn by Chem-Draw 2014. Fluorescent images were collected by Confocal Microscopy (Zeiss Axio-Imager LSM-800). IR thermal images were collected by Infrared Camera (Fotric 255). Photoacoustic imaging data was processed by PA Tomography (Vevo LAZR). Flow cytometry analysis was carried out by FlowJo (version 10.4).

**Reporting summary**. Further information on research design is available in the Nature Research Reporting Summary linked to this article.

## Data availability
The raw data of Figs. 2–6, Supplementary Figs. 1, 2, 5–11, 13–16, 21, 22, 26–28 and 30–34 can be found in the Source Data. Source data are provided with this paper. The authors declare that all other data supporting the findings of this study are within the paper and its Supplementary Information file. Source data are provided with this paper.

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

## Acknowledgements

This paper was partially supported by the National Research Programs of China (2021YFF0701800 Z.L.), National Natural Science Foundation of China (U20A20254 L.C., 52072253 L.C., 21927803 Z.L., and 52032008 Z.L.), Collaborative Innovation Center of Suzhou Nano Science and Technology, the 111 Project, Joint International Research Laboratory of Carbon-Based Functional Materials and Devices, a Jiangsu Natural Science Fund for Distinguished Young Scholars (BK20211544 L.C.) and Jiangsu Social Development Project (BE2019658 Z.L.), a Project Funded for Postgraduate Research and Practice Innovation Program of Jiangsu Province (KYCX21_2947 N.Y.), a China Postdoctoral Science Foundation (2021TQ0229 F.G.), and Suzhou Key Laboratory of Nanotechnology and Biomedicine. L.C. was supported by the Tang Scholarship of Soochow University. We thank the website of app.Biorender.com for the assistance in creating the illustration figures, and we also thank Dr. Anyanee Kamkaew from Suranaree University of Technology for polishing language.

## Author contributions

Z.L. and L.C. oversaw all research; N.Y., L.C., and Z.L. designed the experiments; N.Y., F.G., B. L., Y. H., Y. C., H. L., X. Y., Y. G., and X. W. performed the experiments; all authors analyzed and interpreted the data; N.Y., L.C., and Z.L. wrote and revised the paper; all authors reviewed and edited the paper.

## Competing interests

The authors declare no competing interests.
