## [Peer Review File · Nature Communications]

REVIEWER COMMENTS

Reviewer #1 (Remarks to the Author): Expert in hydrogen gas and cancer therapy

This paper reported a Mg-based galvanic cell composed of Mg rod and Pt nanoparticles for continuous hydrogen therapy by implantable operation. The therapeutic effects were confirmed in diverse tumor models. Additionally, the authors studied the immune response of antitumor induced by the byproduct $Mg(OH)_2$. However, the size of the nanomaterial is too large, which is not fit for in vivo applications. I do not recommend the manuscript for publication in Nature Communications.

1. From this work, it was found that the galvanic cells showed good tumor therapeutic efficacy after multiple intra-tumoral implantation operation. Compared with the traditional surgical resection of tumor, in this work what are the advantages and disadvantages of the implantable H_2 therapy. Please discuss it.
2. I suggest the authors offer more discussion about the current advances of H_2 therapy in the introduction of this article.
3. I suggest the author should provide more detail experiment procedures on how to implant the Mg-based galvanic cells into the tumor, how to fix the cells in tumor. Whether the cells will slip from the tumor during the treatment.
4. Please characterize the diameter of the galvanic cells and the Pt nanoparticles on the Mg rods, and provide the metabolism of the Pt nanoparticles after degradation of Mg in the tumor.
5. I suggest the authors should further investigated the biosafety of the galvanic cells both in vitro and in vivo to confirm their potential clinical applications.
6. Please mark the cellular death ratio of different death forms in the flow cytometry analysis.
7. Normally, H_2 could target and impair the mitochondrion, and destroy the cellular redox balance, leading to cell apoptosis. Moreover, the H&E and TUNEL staining results in Figure 3L displayed significant tumor cell apoptosis after H_2 treatment. But the results of the flow cytometry analysis in Figure 2G showed that the H_2 therapy of the galvanic cells resulted in massive cell necrosis. Please explain it.
8. In this work, one of the most highlight is realization of continuous H_2 supply for H_2 gas treatment. Thus, I suggest the author should add more discussion on how H_2 can kill cancer cells to further clarify the mechanism of H_2 therapy. For example, the changes of intracellular ATP and ROS with time during different treatments.
9. The byproduct $Mg(OH)_2$ could effectively neutralize the acidic tumor environment, and further induce antitumor immune response. Thus, I wonder which of H_2 therapy and immunotherapy works better in this work. This work will be more meaningful, if the authors could compare their therapeutic effects.
10. In order to more intuitively display the therapeutic efficacy in different groups, authors had better offer the tumor photographs after different treatments.
11. There are too many unnecessary references. This makes it practically difficult to check the necessary information.
12. In Figure 1F, the “absorance” should be “absorbance”.

Reviewer #2 (Remarks to the Author): Expert in hydrogen gas and cancer therapy

The Mg-Pt galvanic cell was prepared for hydrogen production and applied in cancer therapy. This method is relatively new, the design is proper for the biomedical application, and the anticancer properties of MgG is well-supported by data in animal models. The overall novelty and significance of this manuscript reaches top 10% compared to most literature published.

- As mentioned in the literature, the anti-cancer treatment of H₂ is dosage-dependent, so a quantitative analysis on MgG therapy is needed.

1. The manuscript claims continuous generation of hydrogen by MgG compared to nanoparticles, but the hydrogen level is only tested for 24 hours in Fig.1H. A quantitative measurement on hydrogen level should be extended to a week (or more) to support the conclusion in animal models.

2. In line 86, the authors reported that no hydrogen bubbles were observed on the Mg rod without using galvanic cells. This piece of information may not be true. A comparative measurement on the Mg-control group should also be conducted, and the obtained data should be added in Fig.1H.

3. Parameters controlling the Galvanic corrosion rate of MgG was not discussed.

- Advantages of using MgG therapy are not illustrated very well.

4. As the degradation of MgG proceeds, would Pt nanoparticles depart from the surface of Mg? Will this affect the continuous generation of hydrogen?

5. Alloying also increase the corrosion rate of Mg in the body and many commercialized Mg alloys are available (Mg-Ca, Mg-Zn, Mg-Mn et al). Are there advantages of using MgG rather than simple Mg alloys in anti-cancer treatment?

- Uncertainly about the immune-modulating roles of MgG therapy

6. Since basis of hydrogen therapy relates to the disruption of redox hemostasis, it is not clear to me why CD8⁺ cells are recruited/activated?

7. Is Mg(OH)₂ pro-inflammatory? Any synergistic effect between Mg(OH)₂ and H₂? Necessary to compare the immune-modulating activity of Mg(OH)₂, H₂, Mg(OH)₂ & H₂.

8. In Fig.3F-H, meaning of the parameter in the x- axis is not clear. What does the number 1,2,3 in the x axis stand for?

Reviewer #3 (Remarks to the Author): Expert in metal-catalyst for hydrogen production

In this work titled “Magnesium Galvanic Cells for Cancer Hydrogen Therapy”, the authors for the first time proposed the idea of designing galvanic cells based on implantable metals for hydrogen therapy, which was quite meaningful. However, there are still some issues needed to be cleared.

1. In this work, the self-assembly of MgG is realized by the difference of metal reduction potential between Mg²⁺/Mg and PtCl₆²⁻/Pt. It is worth noting that the ion concentration (PtCl₆²⁻) and the immersion time will directly affect the loading of the Pt component in the self-assembly process, which may have an impact on the efficiency of hydrogen evolution. Therefore, it is recommended to supplement MgG samples prepared under different reaction time and different PtCl₆²⁻-ion

concentration conditions, and perform corresponding electrochemical analysis to determine its hydrogen evolution performance.

2. It is recommended to supplement the XPS characterization of MgG before and after hydrogen evolution process.

3. Generally, the galvanic reaction couples hydrogen evolution (cathode reaction) and anodic oxidation reaction. Magnesium is oxidized in the anode reaction to form a $Mg(OH)_2$ covering layer. Is it possible that the anodic process also has oxygen evolution behavior? Will the production of oxygen affect the experimental body? It is recommended to test the in-situ gas chromatograph to detect O_2 .

4. In Figure 3L, 4D, 4H, 4L, the cell apoptosis tuningl experiment lacks positive control, and the picture is not clear enough, it is recommended to adjust.

5. It is recommended to adjust 5D image sharpness. Additionally, in the control group of Figure 5D, the kidney H&E chart shows that there are generally large gaps between cells. Is it a pathological injury?

Itemized Responses to Reviewers' Comments

Reviewer #1: (Remarks to the Author): Expert in hydrogen gas and cancer therapy

This paper reported a Mg-based galvanic cell composed of Mg rod and Pt nanoparticles for continuous hydrogen therapy by implantable operation. The therapeutic effects were confirmed in diverse tumor models. Additionally, the authors studied the immune response of antitumor induced by the byproduct Mg(OH)₂. However, the size of the nanomaterial is too large, which is not fit for in vivo applications. I do not recommend the manuscript for publication in Nature Communications.

Reply: Thank the reviewer for these comments. First of all, thank you for highlighting the novelty of this manuscript. The main problem is the bioapplications of the Mg-based galvanic cells with large sizes. Recently, implantable materials and devices (e.g., radioactive seeds) have been widely used in cancer treatments (e.g., liver cancer, and prostate cancer) and have achieved great therapeutic effects. Moreover, the local implantation could avoid the systemic toxicity of therapeutic agents. Local administration of the implantable H₂ therapy can achieve excellent therapeutic effect through minimally invasive methods (such as percutaneous intervention). Moreover, such a strategy based on MgG could achieve a great therapeutic effect without the need to introduce external stimulations nor combine with other treatments, and would be more convenient than the currently developed gas-induced therapies.

1. From this work, it was found that the galvanic cells showed good tumor therapeutic efficacy after multiple intra-tumoral implantation operation. Compared with the traditional surgical resection of tumor, in this work what are the advantages and disadvantages of the implantable H₂ therapy. Please discuss it.

Reply: Thanks for your kind comment. We have discussed the advantages and disadvantages of the implantable H₂ therapy in the revised manuscript. Recently, implantable materials and devices (e.g., radioactive seeds) have been widely used in cancer treatments (e.g., liver cancer, and prostate cancer) and have achieved great therapeutic effects. Moreover, the local implantation could avoid the systemic toxicity of therapeutic agents. Local administration of the implantable H₂ therapy can achieve excellent therapeutic effect through minimally invasive methods (such as percutaneous intervention). However, the implantable H₂ therapy still has some disadvantages, such as the difficulty of operation and intervention for tumors in important organs or complex parts of the body (such as the mediastinum, pancreas, and intracranial tumors).

Revision: Page 3, Paragraph 1, we have added the following sentence.

“Recently, implantable materials and devices (e.g., radioactive seeds) have been widely used in cancer treatments (e.g., liver cancer, and prostate cancer) and have achieved great therapeutic effects. Moreover, the local implantation could avoid the systemic toxicity of therapeutic agents. (24, 25). Local administration of the implantable H₂ therapy can achieve excellent therapeutic effect through minimally invasive methods (such as percutaneous intervention).”

Revision: Page 14, Paragraph 2, we have added the following sentence.

“Local administration of the implantable H₂ therapy can achieve excellent therapeutic effect through minimally invasive methods (such as percutaneous intervention). However, the implantable H₂ therapy still has some disadvantages, such as the difficulty of operation and intervention for tumors in the important organs or complex parts of the body (such as the mediastinum, pancreas, and intracranial tumors).”

Revision: Page 14, Paragraph 2, we have modified the following sentence.

“Importantly, our strategy based on MgG could achieve a great therapeutic effect without the need to introduce external stimulations nor combine with other treatments, and would be more convenient than the currently developed gas-induced therapies.”

Revision: Page 14, Paragraph 2, we have modified the following sentence.

“In addition, cancer immunotherapy has presented tremendous promises by boosting the immune system to fight cancer in recent years (47-50), and the acidic tumor pH is an immunosuppressive feature of solid tumors. The strategy of neutralizing tumor acidity with MgG rods would thus be favorable for the antitumor immune response (40, 51).”

2. *I suggest the authors offer more discussion about the current advances of H₂ therapy in the introduction of this article.*

Reply: Thanks for your good suggestion. We have added the information of the current advances of H₂ therapy in the introduction.

Revision: Page 2, Paragraph 1, we have added the following sentence.

“Currently, H₂ gas has been shown to be effective in treating many diseases, including cancer, type II diabetes mellitus, Parkinson’s disease, Alzheimer’s disease, stroke, and arthritis (7, 8).”

Revision: Page 2, Paragraph 1, we have modified the following sentence.

“It has been found that a low concentration of H₂ could regulate the inflammatory, while H₂ at a high concentration would inhibit cell mitochondrial respiration and break redox homeostasis, thus causing cancer cell apoptosis and damages (2, 5, 12).”

Revision: Page 2, Paragraph 1, we have added the following sentence.

“For instance, palladium hydride (PdH_{0.2}) nanocrystals, SnS_{1.68}-WO_{2.41} nanocatalysts, and Au-TiO₂ heterojunction nanoplateforms (Au-TiO₂@ZnS) have been used to realize in situ H₂ release for tumor gas therapy.”

3. *I suggest the author should provide more detail experiment procedures on how to implant the Mg-based galvanic cells into the tumor, how to fix the cells in tumor. Whether the cells will slip from the tumor during the treatment.*

Reply: Thank the reviewer for this good comment. The Mg-based galvanic cells were implanted into the tumor by a simple percutaneous implantation approach, which was similar to radioactive implanted seeds (Figure S18). During the treatment, the Mg-based galvanic cells would not slip from the tumor. To demonstrate that the galvanic cells remain in the tumor, the *in vivo* time-dependent ultrasonic imaging of 4T1 tumor-bearing mice post intratumoral implantation with MgG rods was conducted (Figure 3A&B). After MgG rods were implanted into the tumor, strong ultrasound signals appeared in the tumor for more than 48 h, suggesting efficient gas generation. To further demonstrate that MgG rods remain in the tumor, SEM images and EDS element mapping of MgG rods after implantation into the tumor for 4 and 24 hours were added (Figure S20). It was found that the galvanic cell structure accelerated the corrosion of Mg to generate enough H₂ gas. The MgG rods were seriously corroded at 24 hours, while Pt nanoparticles still existed on their surface, allowing further H₂ generation for gas therapy. The H₂ generation profiles of MgG rods in PBS solution after implantation into tumor were measured by gas chromatography (Figure S21). Notably, the MgG rods implanted into the

tumor indeed showed efficient H₂ generation at 4 and 24 hours as evidenced by gas chromatography.

Figure S18. Schematic illustration the position of Mg or MgG rods in the tumors of mice (A&B) or rabbits (C&D) implanted by a simple implant approach (E).

Figure S20. SEM image and EDS element mapping of MgG rods after implantation into the tumor for 4 h (A) and 24 h (B).

Figure S21. H₂ generation from MgG rods in PBS solutions after implantation into the tumor for 4 (A) and 24 hours (B) as measured by gas chromatography.

Revision: Page 9, Paragraph 2, we have modified the following sentence.

“For Mg or MgG rods implantation, two Mg or MgG rods (D = 0.5 mm, L = 4 mm) were implanted into each tumor (~7 mm × 7 mm) (Figure S18).”

Revision: Page 11, Paragraph 3, we have modified the following sentence.

“Due to the large rabbit tumor sizes (~800 mm³, 5 times larger than that on mice), three Mg or MgG rods (L = 8.0 mm, D = 0.8 mm) were implanted into each tumor (~12 mm × 12 mm) for the rabbit treatment (Figure S18).”

Revision: Supporting Information, Page 6, Paragraph 1, we have modified the following sentence.

“The mice were implanted with two Mg rods or MgG rods (D = 0.5 mm, L = 4 mm) for each tumor, and the rabbits were implanted with three MgG rods (D = 0.8 mm, L = 8 mm) for each tumor by a simple percutaneous implantation approach (Figure S18).”

Revision: Page 8, Paragraph 2, we have added the following sentence.

“In addition, SEM images and EDS element mapping of MgG rods after implantation into the tumor for 4 and 24 hours showed that the galvanic cell structure accelerated the corrosion of Mg to generate enough H₂. Despite the obvious corrosion of MgG rods, Pt NPs remained on the surface of those rods at 24 hours, allowing further *in vivo* corrosion of MgG and thus the generation of H₂ for gas therapy (Figure S20). Notably, the MgG rods implanted into the tumor indeed showed efficient H₂ generation at 4 and 24 hours as evidenced by gas chromatography (Figure S21).”

4. Please characterize the diameter of the galvanic cells and the Pt nanoparticles on the Mg rods, and provide the metabolism of the Pt nanoparticles after degradation of Mg in the tumor.

Reply: Thanks for your comment. The diameter of galvanic cells is ~0.5 mm, and the Pt NPs (~3 nm) were evenly distributed over Mg rods. In addition, the metabolism of the Pt nanoparticles after degradation from MgG in the tumor and other main organs was measured. The Pt content in the tumor gradually decreased over time, and the major distribution of Pt was detected in the kidney, demonstrating the renal clearance of those detached Pt nanoparticles with ultrasmall sizes detached after etching of Mg rods (Figure 5C&D).

Figure S5. Characterization of Pt nanoparticles (NPs). (A) TEM image of Pt NPs. (B) Particle-size distribution (PSD) of Pt NPs determined by TEM image. (C) High-angle annular dark-field scanning TEM (HAADF-STEM) image and elemental mapping of Pt NPs.

Figure 5. (C) The Pt levels in tumors of mice after being implanted with MgG rods for different periods of time (1, 2, 3, 7, and 15 D). (D) The Pt levels in major organs of mice after being implanted with MgG rods for different periods of time (1, 2, 3, 7, and 15 D).

Revision: Page 5, Paragraph 1, we have modified the following sentence.

“Scanning electron microscopy (SEM) images and energy dispersive spectrometry (EDS) elemental mapping of a MgG rod (~0.5 mm) showed homogeneous distribution of Mg and Pt elements (Figure 1B and Figures S3&S4). Transmission electron microscopy (TEM) image and elemental mapping showed that the Pt nanoparticles (NPs, ~3 nm) were uniformly distributed over the Mg rods (Figure S5).”

Revision: Page 12, Paragraph 2, we have added the following sentence.

“In addition, the biodistribution of Pt NPs after degradation from MgG in the tumor and the main organs was studied. Pt content in the tumor was gradually decreased over time, and the major distribution of Pt was detected in the kidney, indicating the renal clearance of those Pt NPs with their ultrasmall sizes (Figure 5C&D).”

5. I suggest the authors should further investigated the biosafety of the galvanic cells both in vitro and in vivo to confirm their potential clinical applications.

Reply: Thanks for your suggestion. To demonstrate the safety of the constructed galvanic cells, some relevant and important experiments have been carried out in this study. The body weight variation of 4T1 tumor-bearing mice post various treatments was shown in Figure S27. There was no significant body weight loss for mice implanted with MgG rods, indicating no obvious side effects induced by MgG-based H₂ therapy. The Mg levels in tumors and other major organs of mice after being implanted with MgG rods for different periods of time were measured by ICP-OES. Mg contents in the tumor gradually decreased over time, while no notable variation of Mg levels was found in the blood or other organs of the implanted mice compared to the control (Figure 5A&B). Blood biochemistry data of Balb/c mice after being implanted with MgG rods for different periods of time (0, 6 h, 24 h, 48 h, 72 h, 7 days, and 15 days) were also measured (Figure 5F). The measured indexes, including alanine aminotransferase (ALT), aspartate aminotransferase (AST), alkaline phosphatase (ALP), and blood urea, were within the reference range comparable to the control, indicating that the MgG rods implanted caused no significant toxicity. Meanwhile, no obvious histological damage was observed on the main organ of the mice treated with MgG rods (Figure 5E). All the above-related experimental results strongly proved the safety of the constructed Mg galvanic cells.

To further demonstrate the safety of the galvanic cells, more comprehensive safety-related experiments were added. The hematology profiles and blood pH value variation of the mice implanted with MgG rods (D = 0.5 mm, L = 4.0 mm, two rods per mouse) were examined. For the hematological assessment, the white blood cells (WBCs), red blood cells (RBCs), hemoglobin (HGB), hematocrit (HCT), mean corpuscular volume (MCV), mean corpuscular hemoglobin (MCH), mean corpuscular hemoglobin concentration (MCHC), and platelets (PLTs) were selected (Figure S34). All these hematological assay data were found to be normal for MgG-treated mice. Moreover, there was no obvious pH value variation of in the blood of those mice after being implanted with MgG rods (Figure 5G).

In addition to the 4T1 tumor model, there was no significant body weight loss for the CT26 tumor-bearing mice and PDX tumor-bearing mice being implanted with MgG rods, indicating no obvious side effects induced by MgG-based H₂ therapy (Figures S30&32). Importantly, the metabolism of the Pt nanoparticles after degradation from MgG in the tumor and other main organs was measured. The Pt content in the tumor gradually decreased over time, and the major distribution of Pt was detected in the kidney, demonstrating the renal clearance of those detached Pt nanoparticles with ultrasmall sizes detached after etching of Mg rods (Figure 5C&D). Taken together, all these above results illustrated that MgG rods could be rapidly degraded after implantation and displayed no obvious toxicity to the treated mice.

Figure S34. The hematology profiles of mice implanted with MgG rods (D = 0.5 mm, L = 4.0 mm, two rods per mouse) at various time points (0, 3, 14, and 30 D).

Figure 5. (G) Blood pH value variation of Balb/c mice after being implanted with MgG rods for different periods of time (0, 4 h, 12h, 24h, 72 h, and 7 days).

Figure S30. The body weight variation of CT26 tumor-bearing mice post various treatments.

Figure S32. The body weight variation of PDX tumor-bearing mice post various treatments.

Figure 5. (C) The Pt levels in tumors of mice after being implanted with MgG rods for different periods of time (1, 2, 3, 7, and 15 D). (D) The Pt levels in major organs of mice after being implanted with MgG rods for different periods of time (1, 2, 3, 7, and 15 D).

Revision: Page 13, Paragraph 1, we have added the following sentence.

“For the hematological assessment, the white blood cells (WBCs), red blood cells (RBCs), hemoglobin (HGB), hematocrit (HCT), mean corpuscular volume (MCV), mean corpuscular hemoglobin (MCH), mean corpuscular hemoglobin concentration (MCHC), and platelets (PLTs) were selected (Figure S34), and all these hematological assay data were found to be normal in the MgG-treated groups compared with that in the control group.”

Revision: Page 13, Paragraph 1, we have added the following sentence.

“Moreover, there was no obvious pH value variation in blood of Balb/c mice after being implanted with MgG rods (Figure 5G).”

Revision: Page 11, Paragraph 1, we have added the following sentence.

“Additionally, there was no significant body weight loss for the mice implanted with MgG rods, indicating no obvious side effects induced by MgG-based H₂ therapy (Figure S30).”

Revision: Page 11, Paragraph 2, we have added the following sentence.

“Additionally, there was no significant decrease of body weight for the mice being implanted with MgG rods, indicating good safety of MgG-based H₂ therapy (Figure S32).”

Revision: Page 12, Paragraph 2, we have added the following sentence.

“In addition, the biodistribution of Pt NPs after degradation from MgG in the tumor and the main organs was studied. Pt content in the tumor was gradually decreased over time, and the major distribution of Pt was detected in the kidney, indicating the renal clearance of those Pt NPs with their ultrasmall sizes (Figure 5C&D).”

6. Please mark the cellular death ratio of different death forms in the flow cytometry analysis.

Reply: Thanks for your good suggestion. The cellular death ratio of different death forms has been marked in the flow cytometry analysis.

Figure 2. (G) Flow cytometry analysis of 4T1 cells after various treatments using an Annexin V-FITC/PI kit.

Figure S17. Flow cytometry analysis of CT26 cells after various treatments using an Annexin V-FITC/PI kit.

7. Normally, H₂ could target and impair the mitochondrion, and destroy the cellular redox balance, leading to cell apoptosis. Moreover, the H&E and TUNEL staining results in Figure 3L displayed significant tumor cell apoptosis after H₂ treatment. But the results of the flow cytometry analysis in Figure 2G showed that the H₂ therapy of the galvanic cells resulted in massive cell necrosis. Please explain it.

Reply: Thank the reviewer for these insightful comments. As you stated, H₂ could target and impair the mitochondrion, and destroy the cellular redox balance, thus leading to cell apoptosis. The related experiments have been repeated with the multiple cell lines (4T1 and CT26 cells). The cellular death ratios of different death forms were also marked in the flow cytometry analysis.

Figure 2. (G) Flow cytometry analysis of 4T1 cells after various treatments using an Annexin V-FITC/PI kit.

Figure S17. Flow cytometry analysis of CT26 cells after various treatments using an Annexin V-FITC/PI kit.

Revision: Page 7, Paragraph 2, we have modified the following sentence.

“Moreover, the flow cytometry analysis further confirmed that MgG treatment would cause significant apoptosis of 4T1 and CT26 cells (Figure 2G and Figure S17).”

8. In this work, one of the most highlight is realization of continuous H₂ supply for H₂ gas treatment. Thus, I

suggest the author should add more discussion on how H₂ can kill cancer cells to further clarify the mechanism of H₂ therapy. For example, the changes of intracellular ATP and ROS with time during different treatments.

Reply: Thanks for your insightful comments. To further investigate the mechanism of H₂ gas therapy, the changes of intracellular ATP and ROS with time were performed in the revised manuscript. It was found that H₂ generated from MgG drastically altered the oxidative stress state of cancer cells (Figures 2C&D). In addition, the ATP concentrations within 4T1 cells showed a remarkable decrease after being treated with MgG rods (Figure 2E), suggesting that the cellular activity was gradually inhibited by the continuously generated H₂, which thus caused a remarkable reduction in cellular energy.

Figure 2. (C) Time-dependent changes of intracellular ROS in 4T1 cells during different treatments. (D) Flow cytometry data to show DCF-positive 4T1 cells after different treatments.

Figure 2. (E) Time-dependent changes of intracellular ATP contents in 4T1 cells during different treatments.

Revision: Page 7, Paragraph 1, we have modified the following sentence.

“As expected, H₂ generated from MgG rods drastically altered the oxidative stress state of cancer cells with time (Figures 2C&D and Figure S12).”

Revision: Page 7, Paragraph 1, we have modified the following sentence.

“It was found that the ATP concentration within 4T1 cells showed a remarkable decrease over time after being treated with MgG rods (Figure 2E), suggesting that the cellular activity was inhibited by the continuously generated H₂, which thus caused a remarkable reduction in cellular energy.”

Revision: Page 7, Paragraph 2, we have modified the following sentence.

“Moreover, the flow cytometry analysis further confirmed that MgG treatment would cause significant

apoptosis of 4T1 and CT26 cells (Figure 2G and Figure S17)."

9. The byproduct $Mg(OH)_2$ could effectively neutralize the acidic tumor environment, and further induce antitumor immune response. Thus, I wonder which of H_2 therapy and immunotherapy works better in this work. This work will be more meaningful, if the authors could compare their therapeutic effects.

Reply: Thanks for your comment. Solid tumors are characterized by an acidic tumor microenvironment, which may impede effective antitumor T-cell immune responses. Specifically, $CD8^+$ T cells tend to become anergic and apoptotic when exposed to a low pH environment. The neutralization of tumor acidity would be favorable for antitumor immune responses. In this work, the by-product $Mg(OH)_2$ could modulate the immunosuppressive TME for improved tumor therapy. It could be found that the Mg-implanted groups showed no significant tumor-suppressing effect (Figures 3J-M). The tumors of $Mg(OH)_2$ group showed slightly delayed growth, due to the neutralization of tumor pH. However, the tumor growth of MgG rods implantation group was significantly inhibited.

Therefore, H_2 played a major killing role in this process, and the by-product $Mg(OH)_2$ would mainly regulate the immune microenvironment to enhance tumor treatment. To better illustrate this mechanism, we added a $Mg(OH)_2$ control group and repeated tumor immunoassay. Compared with other control groups, the $Mg(OH)_2$ treated group showed a slight regulatory effect, while the tumors from the MgG group exhibited reduced populations of MDSCs and the increased percentage of total T cells (especially $CD8^+$ T cells). MDSCs, as immunosuppressive cells, could promote tumor progression by inhibiting anti-tumor immunity, while T cells (especially $CD8^+$ T cells) would ensure effective tumor cell killing. Therefore, MgG implantation would thus modulate the immunosuppressive TME for the enhanced tumor therapy (Figures 3F-I, and Figures S23&24).

Figure 3. (F-H) The quantification results of MDSCs ($CD45^+CD11b^+Gr-1^+$, F), T cells ($CD3^+$, G), $CD8^+$ T cells ($CD3^+CD8^+$, H) by flow cytometry on day 6 post MgG implantation. (I) The flow cytometric analysis results of $CD8^+$ T cells ($CD3^+CD8^+$) within the tumors after different treatments.

Figure S23. The flow cytometric analysis results of T cells (CD3⁺) within the tumors after different treatments.

Figure S24. The flow cytometric analysis results of myeloid-derived suppressor cells (MDSCs, CD45⁺CD11b⁺Gr-1⁺) within the tumors after different treatments.

Revision: Page 9, Paragraph 1, we have modified the following sentence.

“Solid tumors are characterized by an acidic microenvironment, which may impede effective antitumor T-cell immune responses. Specifically, CD8⁺ T cells tend to become anergic and apoptotic when exposed to a low pH environment. The neutralization of tumor acidity would be favorable for antitumor immune responses (40, 41).”

Revision: Page 9, Paragraph 1, we have modified the following sentence.

“Compared with the other groups, the Mg(OH)₂ group showed a slight regulatory effect, while the tumors from the MgG group exhibited reduced populations of MDSCs and an increased percentage of total T cells (especially CD8⁺ T cells). MDSCs, as immunosuppressive cells, could promote tumor progression by inhibiting anti-tumor immunity, while T cells (especially CD8⁺ T cells) would ensure effective tumor cell killing. Therefore, MgG implantation would thus modulate the immunosuppressive TME for enhanced tumor therapy (Figures 3F-I, and Figures S23&24).”

10. *In order to more intuitively display the therapeutic efficacy in different groups, authors had better offer the tumor photographs after different treatments.*

Reply: Thanks for your suggestion. We have added the related data to intuitively display the therapeutic efficacy. Mice bearing subcutaneous 4T1 tumors expressing firefly luciferase (Luc-4T1) were used to intuitively display the therapeutic efficacy. MgG rods implantation group showed the weakest bioluminescence signals, and demonstrated the best tumor growth inhibition (Figure 3M). In addition, photos of representative mice taken at different days after various treatments also intuitively displayed the great therapeutic efficacy of MgG induced H₂ therapy (Figure S29).

Figure 3. (M) *In vivo* bioluminescence images of mice bearing subcutaneous 4T1 tumors expressing firefly luciferase (Luc-4T1) to display the therapeutic efficacy of mice after various treatments.

Figure S29. Photos of the representative CT26 tumor-bearing mice taken at different days after various treatments.

Revision: Page 10, Paragraph 1, we have added the following sentence.

“Moreover, mice bearing subcutaneous 4T1 tumors expressing firefly luciferase (Luc-4T1) were also used to intuitively display the therapeutic efficacy. MgG rods implantation group showed the weakest bioluminescence signal, and demonstrated the best tumor growth inhibition (Figure 3M).”

Revision: Page 10, Paragraph 2, we have modified the following sentence.

“As expected, MgG rods implantation significantly inhibited the tumor growth, with ~ 40% of tumors being completely eliminated after three times of MgG implantations (Figures 4B&C and Figures S28&29).”

11. There are too many unnecessary references. This makes it practically difficult to check the necessary information.

Reply: Thanks for your kind reminder. We have modified the related references.

12. In Figure 1F, the “absorance” should be “absorbance”.

Reply: Thanks for your kind reminder. We have modified the Figure 1F and checked the whole manuscript.

Figure 1F Time-dependent absorption spectra of the MB solution (pH = 6.5) with MgG rods added.

Reviewer #2 (Remarks to the Author): Expert in hydrogen gas and cancer therapy

The Mg-Pt galvanic cell was prepared for hydrogen production and applied in cancer therapy. This method is relatively new, the design is proper for the biomedical application, and the anticancer properties of MgG is well-supported by data in animal models. The overall novelty and significance of this manuscript reaches top 10% compared to most literature published.

Reply: Thank the reviewer for this positive comment. We have carefully revised the manuscript according to your comments.

•As mentioned in the literature, the anti-cancer treatment of H_2 is dosage-dependent, so a quantitative analysis on MgG therapy is needed.

1.The manuscript claims continuous generation of hydrogen by MgG compared to nanoparticles, but the

hydrogen level is only tested for 24 hours in Fig.1H. A quantitative measurement on hydrogen level should be extended to a week (or more) to support the conclusion in animal models.

Reply: Thanks for your comment. A quantitative measurement on hydrogen generation level has been extended to a week by gas chromatography (Figure 1H). It found that MgG rods (PtCl₆²⁻ concentration: 0.3%, immersion time: 1 min) were able to continuously generate H₂ gas about 160 μmol for more than 1 week.

Figure 1. (H) Time-dependent H₂ generation measured by gas chromatography from Mg or MgG rods in PBS (pH = 6.5).

Revision: Page 5, Paragraph 2, we have modified the following sentence.

“Quantitative measurement of H₂ generation from the MgG and Mg rods was further conducted by gas chromatography. It was found that compared to bare Mg rods, MgG rods were able to generate more H₂ gas, continuously for more than one week (Figure 1H).”

2. In line 86, the authors reported that no hydrogen bubbles were observed on the Mg rod without using galvanic cells. This piece of information may not be true. A comparative measurement on the Mg-control group should also be conducted, and the obtained data should be added in Fig.1H.

Reply: Thanks for your insightful comments. Indeed, by immersing into water, MgG rods could rapidly react with H₂O and generate a large amount of hydrogen gas bubbles, while bare Mg rods virtually generated no obvious bubbles. A quantitative measurement on hydrogen generation level of Mg-control group was also performed. It was found that compared to bare Mg rods, MgG rods were able to generate more H₂ gas, continuously for more than one week.

Figure 1. (H) Time-dependent H₂ generation measured by gas chromatography from Mg or MgG rods in PBS (pH = 6.5).

Revision: Page 5, Paragraph 2, we have modified the following sentence.

“Indeed, by immersion in water, MgG rods could rapidly react with H₂O and generate lots of H₂ gas bubbles, while bare Mg rods virtually generated no obvious bubbles (Figure 1D).”

Revision: Page 5, Paragraph 2, we have modified the following sentence.

“Quantitative measurement of H₂ generation from the MgG and Mg rods was further conducted by gas chromatography. It was found that compared to bare Mg rods, MgG rods were able to generate more H₂ gas, continuously for more than one week (Figure 1H).”

3. *Parameters controlling the Galvanic corrosion rate of MgG was not discussed.*

Reply: Thanks for your comment. To optimize the MgG parameters, the hydrogen generation performance of MgG prepared under different reaction durations and concentrations of PtCl₆²⁻ were performed using gas chromatography. It was found that the H₂ generation performance was enhanced with increasing ion concentrations and immersion time, and then reached to a steady-state.

Figure S1. H₂ generation in 24 h from MgG rods prepared under different PtCl₆²⁻ concentrations as measured by gas chromatography.

Figure S2. H₂ generation in 24 h from MgG rods prepared under different immersion time as measured by gas chromatography.

Revision: Page 4, Paragraph 2, we have added the following sentence.

“It is known that the ion concentration (PtCl₆²⁻) and the immersion time directly affect the Pt loading component via the self-assembly process, which may have an impact on the efficiency of H₂ generation. To optimize the MgG parameters, the H₂ generation performance of MgG rods prepared under different immersion time and concentrations of PtCl₆²⁻ was determined using gas chromatography. The H₂ generation performance was enhanced with increasing ion concentrations and immersion time, and then reached to a steady-state (Figures S1&S2).”

• *Advantages of using MgG therapy are not illustrated very well.*

4. *As the degradation of MgG proceeds, would Pt nanoparticles depart from the surface of Mg? Will this affect*

the continuous generation of hydrogen?

Reply: Thanks for your important comments. As the degradation of MgG proceeds, Pt nanoparticles may partially depart from the surface of Mg, while the Pt signal can still be detected on the surface of Mg after 24 hours. To demonstrate that the galvanic cells remain in the tumor, the *in vivo* time-dependent ultrasonic imaging of 4T1 tumor-bearing mice post intratumoral implantation with MgG rods has been given in the original manuscript (Figure 4A). In addition, the therapeutic efficacy also demonstrated that the galvanic cells remain in the tumor to generate H₂ bubbles for hydrogen therapy. To demonstrate that MgG rods remain in the tumor, SEM images and EDS element mapping of MgG rods after implantation into the tumor for 4 and 24 hours were added (Figure S20). It was found that the galvanic cell structure accelerated the corrosion of Mg to generate more H₂ gas. The MgG rods were seriously corroded at 24 hours, while Pt NPs still existed on its surface, which was convenient for the further H₂ generation for gas therapy. The H₂ generation profiles of MgG rods in PBS solutions after implantation into the tumor were measured by the gas chromatography (Figure S21). Notably, the MgG rods implanted into the tumor indeed showed efficient H₂ generation at 4 and 24 hours as evidenced by gas chromatography.

Figure S20. SEM image and EDS element mapping of MgG rods after implantation into the tumor for 4 h (A) and 24 h (B).

Figure S21. H₂ generation from MgG rods in PBS solutions after implantation into the tumor for 4 (A) and 24 hours (B) as measured by gas chromatography.

Revision: Page 8, Paragraph 2, we have modified the following sentence.

“In addition, SEM images and EDS element mapping of MgG rods after implantation into the tumor for 4 and 24 hours showed that the galvanic cell structure accelerated the corrosion of Mg to generate enough H₂. Despite the obvious corrosion of MgG rods, Pt NPs remained on the surface of those rods at 24 hours, allowing further *in vivo* corrosion of MgG and thus the generation of H₂ for gas therapy (Figure S20). Notably, the MgG rods implanted into the tumor indeed showed efficient H₂ generation at 4 and 24 hours as evidenced by gas chromatography (Figure S21).”

5. Alloying also increase the corrosion rate of Mg in the body and many commercialized Mg alloys are available (Mg-Ca, Mg-Zn, Mg-Mn et al). Are there advantages of using MgG rather than simple Mg alloys in anti-cancer treatment?

Reply: Thanks for your kind comments. Since alloying may increase the corrosion rate of Mg, the H₂ generation performance and cell-killing effect of MgG rods and other commercialized Mg alloys (Mg, MgZnCa, MgAl) were evaluated. The MgG rods exhibited the strongest cell-killing effect, probably due to their superior H₂ generation capacity,

Figure S14. H₂ generation performance of MgG rods and different kinds of the commercialized Mg alloys (Mg, MgZnCa, and MgAl) measured by gas chromatography.

Figure S15. Relative viabilities of 4T1 cells after various treatments (Control, Mg, MgZnCa, MgAl, and MgG).

Revision: Page 7, Paragraph 2, we have modified the following sentence.

“Since alloying may increase the corrosion rate of Mg, the H₂ generation performance and cell-killing effect of MgG rods and other commercialized Mg alloys (Mg, MgZnCa, MgAl) were evaluated. The MgG rods exhibited the strongest cell-killing effect, probably due to their superior H₂ generation capacity (Figures S14&S15).”

• *Uncertainly about the immune-modulating roles of MgG therapy*

6. *Since basis of hydrogen therapy relates to the disruption of redox hemostasis, it is not clear to me why CD8⁺ cells are recruited/activated?*

Reply: Thanks for your comment. In this manuscript, H₂ gas played a major killing role in this process, and the by-product Mg(OH)₂ was mainly regulated immune microenvironment to enhance tumor treatment. The solid tumors are characterized by an acidic microenvironment, which may impede effective antitumor T-cell immune responses. Specifically, CD8⁺ T cells tend to become anergic and apoptotic when exposed to a low pH environment. The neutralization of tumor acidity would be favorable for antitumor immune responses. Compared with the other groups, the Mg(OH)₂ group showed a slight regulatory effect, while the tumors from the MgG group (producing Mg(OH)₂ continuously) exhibited the reduced populations of myeloid-derived suppressor cells (MDSCs) and the increased percentage of total T cells (especially CD8⁺ T cells). Because MDSCs as immunosuppressive cells could promote tumor progression by inhibiting anti-tumor immunity while T cells (especially CD8⁺ T cells) would ensure effective tumor cell killing, MgG implantation would thus modulate the immunosuppressive TME for the improved tumor therapy.

Revision: Page 9, Paragraph 1, we have modified the following sentence.

“The solid tumors are characterized by an acidic microenvironment, which may impede effective antitumor T-cell immune responses. Specifically, CD8⁺ T cells tend to become anergic and apoptotic when exposed to a low pH environment. The neutralization of tumor acidity would be favorable for antitumor immune responses (40, 41).”

Revision: Page 9, Paragraph 1, we have modified the following sentence.

“Compared with the other groups, the Mg(OH)₂ group showed a slight regulatory effect, while the tumors from the MgG group exhibited reduced populations of MDSCs and an increased percentage of total T cells (especially CD8⁺ T cells). MDSCs, as immunosuppressive cells, could promote tumor progression by inhibiting anti-tumor immunity, while T cells (especially CD8⁺ T cells) ensure effective tumor cell killing. Therefore, MgG implantation would modulate the immunosuppressive TME for enhanced tumor therapy (Figures 3F-I, and Figures S23&24).”

7. *Is Mg(OH)₂ pro-inflammatory? Any synergistic effect between Mg(OH)₂ and H₂? Necessary to compare the immune-modulating activity of Mg(OH)₂, H₂, Mg(OH)₂ & H₂.*

Reply: Thanks for your kind comments. In this work, the by-product Mg(OH)₂ could modulate the immunosuppressive TME for improved tumor therapy. The solid tumors are characterized by an acidic microenvironment, which may impede effective antitumor T-cell immune responses. The neutralization of tumor acidity would be favorable for antitumor immune responses. From the treatment results, Mg(OH)₂ and H₂ showed a synergistic effect to some extent. Mg-implanted groups had no significant tumor-suppressing effect (Figure 3J-M). Treatment with Mg(OH)₂ could slightly delay the tumor growth, likely due to the neutralization of tumor pH value. Remarkably, the growth of tumors with MgG rods implantation was

significantly inhibited. Therefore, in this treatment process, H_2 played a major killing role, and the by-product $Mg(OH)_2$ would regulate the immunosuppressive tumor microenvironment to enhance tumor treatment.

To better illustrate this, we added a $Mg(OH)_2$ control group and repeated tumor immunoassay. Compared with the other groups, the $Mg(OH)_2$ group showed a slight regulatory effect, while the tumors from the MgG group (continuously producing $Mg(OH)_2$) exhibited reduced populations of MDSCs and the increased percentage of total T cells (especially $CD8^+$ T cells). MDSCs, as immunosuppressive cells could promote tumor progression via inhibiting anti-tumor immunity, while T cells (especially $CD8^+$ T cells) would ensure effective tumor cell killing, MgG implantation would thus modulate the immunosuppressive TME for enhanced tumor therapy (Figures 3F-I, and Figures S23&24).

Figure 3. (F-H) The quantification results of MDSCs ($CD45^+CD11b^+Gr-1^+$, F), T cells ($CD3^+$, G), $CD8^+$ T cells ($CD3^+CD8^+$, H) by flow cytometry on day 6 post MgG implantation. (I) The flow cytometric analysis results of $CD8^+$ T cells ($CD3^+CD8^+$) within the tumors after different treatments.

Figure S23. The flow cytometric analysis results of T cells ($CD3^+$) within the tumors after different treatments.

Figure S24. The flow cytometric analysis results of myeloid-derived suppressor cells (MDSCs, CD45⁺CD11b⁺Gr-1⁺) within the tumors after different treatments.

8. In Fig.3F-H, meaning of the parameter in the x-axis is not clear. What does the number 1,2,3 in the x axis stand for?

Reply: Thanks for your kind reminder. The meaning of number 1,2,3,4 in the x axis was added after complementing Mg(OH)₂ control group and repeating tumor immunoassay.

Figure 3. (F-H) The quantification results of MDSCs (CD45⁺CD11b⁺Gr-1⁺, F), T cells (CD3⁺, G), CD8⁺ T cells (CD3⁺CD8⁺, H) by flow cytometry on day 6 post MgG implantation.

Reviewer #3 (Remarks to the Author): Expert in metal-catalyst for hydrogen production

In this work titled "Magnesium Galvanic Cells for Cancer Hydrogen Therapy", the authors for the first time proposed the idea of designing galvanic cells based on implantable metals for hydrogen therapy, which was quite meaningful. However, there are still some issues needed to be cleared.

Reply: Thank the reviewer for this positive comment. We have carefully revised the manuscript according to your comments, and the quality are greatly improved.

1. In this work, the self-assembly of MgG is realized by the difference of metal reduction potential between Mg²⁺/Mg and PtCl₆²⁻/Pt. It is worth noting that the ion concentration (PtCl₆²⁻) and the immersion time will directly affect the loading of the Pt component in the self-assembly process, which may have an impact on the efficiency of hydrogen evolution. Therefore, it is recommended to supplement MgG samples prepared under different reaction time and different PtCl₆²⁻ ion concentration conditions, and perform corresponding electrochemical analysis to determine its hydrogen evolution performance.

Reply: Thanks for your comment. In order to optimize the MgG parameters, the hydrogen generation performance of MgG prepared under different reaction durations and concentrations of PtCl₆²⁻ were performed using gas chromatography. It found that H₂ generation performance was enhanced with increasing ion concentration and immersion time, and then reached to a steady-state. Therefore, the MgG (PtCl₆²⁻ concentration: 0.3%, immersion time: 1 min) were constructed for the further use.

Figure S1. H₂ generation in 24 h from MgG rods prepared under different PtCl₆²⁻ concentrations as measured by gas chromatography.

Figure S2. H₂ generation in 24 h from MgG rods prepared under different immersion time as measured by gas chromatography.

Revision: Page 4, Paragraph 2, we have added the following sentence.

“It is known that the ion concentration (PtCl₆²⁻) and the immersion time directly affect the Pt loading component via the self-assembly process, which may have an impact on the efficiency of H₂ generation. To optimize the MgG parameters, the H₂ generation performance of MgG rods prepared under different immersion time and concentrations of PtCl₆²⁻ was determined using gas chromatography. The H₂ generation performance was enhanced with increasing ion concentrations and immersion time, and then reached to a steady-state (Figures S1&S2).”

2. It is recommended to supplement the XPS characterization of MgG before and after hydrogen evolution process.

Reply: Thanks for your kind comment. The XPS analysis of MgG before and after the hydrogen evolution process was added. It showed strong signals from Mg (II), further proving that MgG rods were sacrificed themselves to generate H₂.

Figure S10. XPS characterization of MgG before (A&B) and after (C&D) H₂ generation process.

Revision: Page 6, Paragraph 1, we have added the following sentence.

“In addition, the XPS spectra of MgG after the H₂ generation process showed strong signals from Mg (II), further proving that MgG rods were sacrificed during the generation of H₂ (Figure S10).”

3. Generally, the galvanic reaction couples hydrogen evolution (cathode reaction) and anodic oxidation reaction. Magnesium is oxidized in the anode reaction to form a Mg(OH)₂ covering layer. Is it possible that the anodic process also has oxygen evolution behavior? Will the production of oxygen affect the experimental body? It is recommended to test the in-situ gas chromatograph to detect O₂.

Reply: Thanks for your kind comment. As you stated, the galvanic reaction couples hydrogen evolution (cathode reaction) and anodic oxidation reaction. Electrons (e⁻) would flow out of the negative electrode (Mg electrode, Mg-2e⁻=Mg²⁺) and flow into the positive electrode (Pt electrode, 2H₂O+2e⁻=2OH⁻+H₂), leading to the water etching of Mg and the H₂ gas generation. During the H₂ generation process, no significant O₂ was generated because Mg was oxidized in the negative electrode reaction. Furthermore, the oxygen generation would not affect the experimental body, since strategies of ameliorating hypoxia are widely used for enhanced cancer therapy.

Figure S9. O₂ content changes after the reaction between Mg or MgG rods and H₂O measured by gas chromatography.

Revision: Page 6, Paragraph 1, we have added the following sentence.

“During the H₂ generation process, no significant O₂ was generated because Mg was oxidized in the negative electrode reaction (Figure S9).”

4. In Figure 3L, 4D, 4H, 4L, the cell apoptosis tuningl experiment lacks positive control, and the picture is not clear enough, it is recommended to adjust.

Reply: Thanks for your kind remind and suggestion. In Figure 3L, 4D, 4H, 4L, the microscopy images of TUNEL stained with the positive controls were given, and the image sharpness was adjusted.

Figure S25. Microscopy images of TUNEL stained with the positive controls in different tumor sections.

Figure 3. (L) Microscopy images of H&E and TUNEL stained tumor slices collected from mice post different treatment groups.

Figure 4. (D) Microscopy images of H&E and TUNEL stained CT26 tumor slices collected from different groups.

Figure 4. (H) Microscopy images of H&E and TUNEL stained PDX tumor slices collected from different groups.

Figure 4. (L) Microscopy images of H&E and TUNEL stained VX₂ tumor slices collected from different groups.

5. It is recommended to adjust 5D image sharpness. Additionally, in the control group of Figure 5D, the kidney H&E chart shows that there are generally large gaps between cells. Is it a pathological injury?

Reply: Thanks for your kind reminder and suggestion. The image sharpness of figure 5D has been adjusted. The gaps between cells in the kidney H&E chart mainly come from the renal pelvis, not the pathological injury. Microscopy H&E images of the kidney were modified in the revised manuscript.

Figure 5. (E) Hematoxylin and eosin (H&E) staining of mouse major organs (liver, spleen, kidney, heart, and lung) to examine the histological changes after implantation of MgG rods in mice.

 We thank the above three reviewers for his/her insightful comments and suggestions, which greatly helped to improve the novelty and quality of our manuscript!

REVIEWERS' COMMENTS

Reviewer #1 (Remarks to the Author):

I suggest to accepting the revised manuscript.

Reviewer #2 (Remarks to the Author):

The revisions have clarified my uncertainty. I'd like to see the paper to be published.

A recently published work showing that the tumor apoptosis rate correlates to the hydrogen-emitting rate from Mg implants, I suggest citing this paper as well.

"Zan R, Wang H, Cai W, et al. Controlled release of hydrogen by implantation of magnesium induces P53-mediated tumor cells apoptosis. *Bioactive materials*, 2022, 9: 385-396."

Reviewer #3 (Remarks to the Author):

The authors have made a good revision, which has satisfied my concern. I suggest to accept as it is.

Itemized Responses to Reviewers' Comments

Reviewer #1:

I suggest to accepting the revised manuscript.

Reply: Thank the reviewer for this positive comment.

Reviewer #2:

The revisions have clarified my uncertainty. I'd like to see the paper to be published.

A recently published work showing that the tumor apoptosis rate correlates to the hydrogen-emitting rate from Mg implants, I suggest citing this paper as well.

"Zan R, Wang H, Cai W, et al. Controlled release of hydrogen by implantation of magnesium induces P53-mediated tumor cells apoptosis. Bioactive materials, 2022, 9: 385-396."

Reply: Thank the reviewer for this positive comment. We have studied and added the related references in the revised manuscript.

Reviewer #3:

The authors have made a good revision, which has satisfied my concern. I suggest to accpet as it is.

Reply: Thank the reviewer for this positive comment.

We thank the above three reviewers for his/her insightful comments and suggestions again, which greatly helped to improve the novelty and quality of our manuscript!